# Gap junction-mediated contraction of myoepithelial cells induces the peristaltic transport of sweat in human eccrine glands

Kie Nakashima [1], Hiroko Kato[1], Ryuichiro Kurata[1], Luo Qianwen[1], Tomohisa Hayakawa[1], Fumihiro Okada[2], Fumitaka Fujita[1,2 ✉], Yukinobu Nakagawa[3], Atsushi Tanemura [3], Hiroyuki Murota[4], Ichiro Katayama[5] & Kiyotoshi Sekiguchi [6 ✉]

Eccrine sweat glands play an essential role in regulating body temperature. Sweat is produced in the coiled secretory portion of the gland, which is surrounded by obliquely aligned myoepithelial cells; the sweat is then peristaltically transported to the skin surface. Myoepithelial cells are contractile and have been implicated in sweat transport, but how myoepithelial cells contract and transport sweat remains unexplored. Here, we perform ex vivo live imaging of an isolated human eccrine gland and demonstrate that cholinergic stimulation induces dynamic contractile motion of the coiled secretory duct that is driven by gap junction-mediated contraction of myoepithelial cells. The contraction of the secretory duct occurs segmentally, and it is most prominent in the region surrounded by nerve fibers, followed by distension-contraction sequences of the excretory duct. Overall, our ex vivo live imaging approach provides evidence of the contractile function of myoepithelial cells in peristaltic sweat secretion from human eccrine glands.

[1] Laboratory of Advanced Cosmetic Science, Graduate School of Pharmaceutical Sciences, Osaka University, Osaka, Japan. [2] Fundamental Research Institute, Mandom Corporation, Osaka, Japan. [3] Department of Dermatology, Graduate School of Medicine, Osaka University, Osaka, Japan. [4] Department of Dermatology, Graduate School of Biomedical Sciences, Nagasaki University, Nagasaki, Japan. [5] Department of Dermatology, Graduate School of Medicine, Osaka City University, Osaka, Japan. [6] Division of Matrixome Research and Application, Institute for Protein Research, Osaka University, Osaka, Japan. ✉email: fujita-f@phs.osaka-u.ac.jp; sekiguch@protein.osaka-u.ac.jp

Humans have two types of sweat glands: eccrine and apocrine. Eccrine glands are essential thermoregulatory organs that secrete a watery fluid, i.e., sweat, while apocrine glands are evolutionary remnants of odor-producing organs that secrete thick, milky mucus[1]. Sweat from eccrine glands removes excess body heat from the skin surface through evaporative cooling. Eccrine gland dysfunction can cause either excessive or inadequate sweat secretion, commonly known as hyperhidrosis and hypohidrosis, respectively. Hyperhidrosis leads to physical discomfort, social stigmatization and even mental stress, whereas hypohidrosis can result in heatstroke, a form of life-threatening hyperthermia[2]. These abnormalities in sweat production are caused by failures in either the hypothalamic thermoregulatory center or local sweat glands[3,4]. Because the hypothalamus is a multimodal regulatory center, understanding the local mechanisms of sweat secretion in human skin is fundamental to developing site-specific treatments for sweating dysfunction.

The eccrine gland forms an unbranched, highly coiled tubule that extends into the deep dermis (Fig. 1a). The coiled portion of the eccrine gland consists of the secretory duct and a transitional region of the excretory duct. The excretory duct is formed by two layers of stratified cuboidal cells, i.e., the luminal and basal layers, while the secretory duct is composed of a monolayer of secretory luminal cells obliquely surrounded by spindle-shaped myoepithelial cells[5] (Fig. 1b). Myoepithelial cells are filled with dense masses of myofilaments comprising smooth muscle actin[5,6], indicative of their ability to contract. Indeed, myoepithelial cells dissociated from the eccrine gland of primate palms are induced to contract upon cholinergic stimulation[7], although the contraction of myoepithelial cells in situ and its relevance to sweat secretion have remained unexplored. The myoepithelial cells in the eccrine gland are arranged longitudinally around the entangled secretory duct and partially surrounded by autonomic nerve fibers in a bandage-like manner[5], suggesting that the contraction of myoepithelial cells may be dominated by synchronous and directional neuronal activities to compress the secretory duct for sweat transport. Therefore, it is necessary to capture the dynamics of the contraction of myoepithelial cells as well as the changes in the lumen volume in secretory and excretory ducts to elucidate the in situ function of myoepithelial cells in eccrine glands.

To demonstrate the contractile motion of the secretory ducts of eccrine glands, we established an ex vivo method of 3D live-tissue imaging of isolated secretory ducts that includes the simultaneous labeling of actin and the cell membrane of both luminal and myoepithelial cells. Time-lapse imaging allowed quantitative visualization of a series of contractile motions of the secretory ducts triggered by pharmacological stimulation. Moreover, 3D analysis of myoepithelial and luminal cells at a single-cell resolution corroborated the essential role of myoepithelial cells in the contractile motions of secretory ducts. We also demonstrated that cholinergic stimulation triggers the contraction of myoepithelial cells to compress secretory ducts and that the propagation of contraction in the myoepithelium requires cell-to-cell communication via gap junctions that are associated with sweat-expelling motions of the secretory duct; thus, we provide technological and conceptual advances for elucidating the molecular mechanisms underlying sweating in humans.

## Results

### 3D live-tissue imaging of myoepithelial cells in isolated eccrine glands.
The secretory duct of the sweat gland is difficult to observe in vivo and ex vivo because the secretory duct, which forms the coiled portion of the sweat gland, is localized in the deep dermis. To visualize the shape and location of myoepithelial cells of the secretory duct, we harvested eccrine glands from human skin and live-stained the whole eccrine coiled structure. Because myoepithelial cells contain abundant filamentous actin, we used Acti-stain 488 (Acti-stain), a fluorochrome-conjugated phalloidin that directly binds to filamentous actin. The staining patterns of Acti-stain for excretory and secretory ducts were distinguishable (Fig. 2a, b). Acti-stain signals were detected in submembranous regions of cuboidal cells in the excretory duct (Fig. 2c) and ubiquitously within spindle-shaped cells in the secretory duct (Fig. 2d). The Acti-stain signals in the spindle-shaped cells overlapped with the signals of α-smooth muscle actin (αSMA), a marker of myoepithelial cells (Fig. 2e, f), indicating that the spindle-shaped cells were myoepithelial cells longitudinally elongated along an entangled secretory tubule. Consistent with a previous report[5], the cuboidal cells in the excretory duct lacked an αSMA signal (Fig. 2g). Furthermore, counterstaining with Hoechst 33342 revealed that nuclei in the excretory duct were densely packed because of compactly aligned luminal cuboidal cells (Fig. 2h, i; arrows), whereas nuclei in the secretory duct were rather sparse because of the elongated myoepithelial cells (Fig. 2h, i; arrowheads). In addition, the nuclei of the myoepithelial cells were typically elliptical (Fig. 2i, arrowheads in the right panel), while those of the luminal cuboidal cells were rounded (Fig. 2i, arrows in the right panel). These results indicate that myoepithelial cells are distinguishable from other cell types based on their cellular and nuclear morphologies on 3D live-tissue imaging.

### Cholinergic stimulation triggers eccrine gland contraction.
Considering that nerve fibers wrap around the secretory duct in a bundle[5], we hypothesized that the myoepithelium in the secretory duct would synchronously contract by autonomic nerve stimulation to extrude sweat from the secretory duct. Thus, we attempted to record the motions of the isolated eccrine glands after labeling their nuclei and actin cytoskeletons with Hoechst 33342 and Acti-stain, respectively. Eccrine glands are innervated by atypical sympathetic terminals, which release acetylcholine[8]. We tracked the motions of both the nuclei and the tissue structure of the coiled portion of the eccrine gland upon stimulation with pilocarpine, a cholinergic agonist, to mimic autonomic nervous stimuli for sweating[9]. To quantitatively describe the

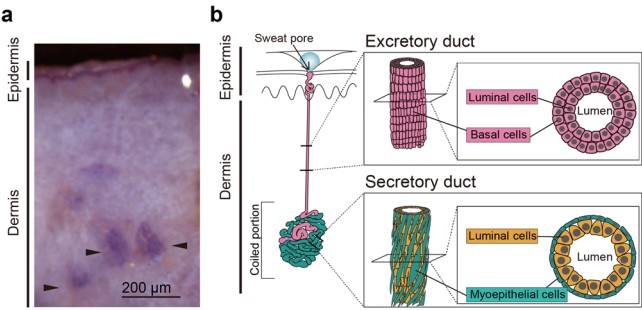

**Fig. 1 Schematic illustration of a human eccrine gland. a** Human eccrine glands live-stained in situ with neutral red. Arrowheads indicate the coiled portions of the eccrine glands. **b** Schematic illustration of a human eccrine gland. The human eccrine gland is an unbranched, coiled single-tubular exocrine gland (left). The coiled structure, which resides in the deep dermis, consists of the secretory duct (SD; green) and the excretory duct (ED; pink). The excretory duct starts within the coiled portion and then extends straight up to the skin surface. The wall of the excretory duct is composed of two layers of cuboidal luminal and basal cells (pink in upper right box). The wall of the secretory duct is composed of a monolayer of secretory luminal cells (orange in lower right box) and the surrounding monolayer of myoepithelial cells (green in lower right box).

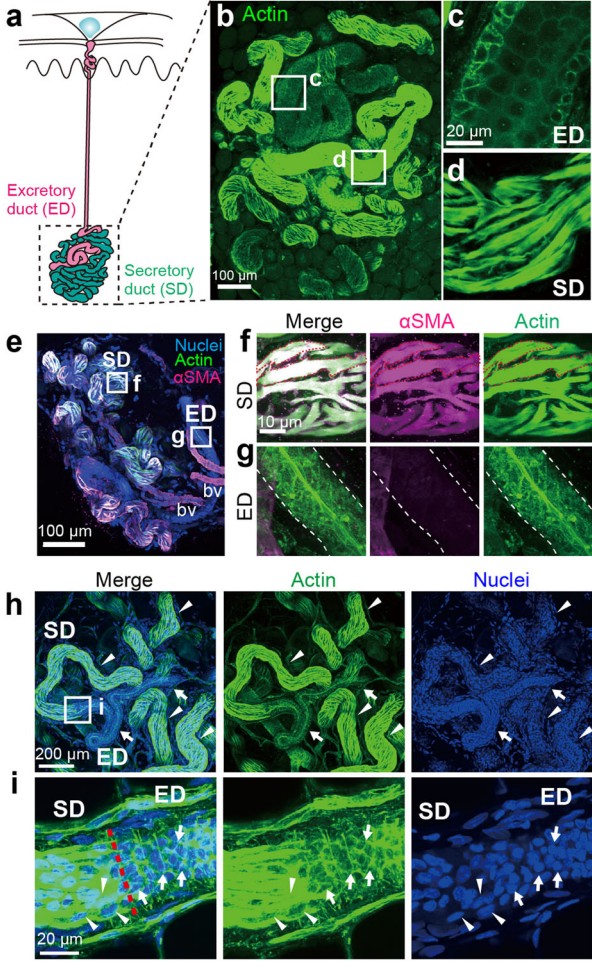

**Fig. 2 Myoepithelial cells in eccrine glands visualized by actin staining.**
**a** Schematic illustration of an eccrine gland. **b** The coiled structure of a human eccrine gland visualized by Acti-stain. High-magnification images of the excretory duct (ED; **c**) and the secretory duct (SD; **d**) in **b**. Acti-stain distinctly delineated the cuboidal cells of the excretory duct (**c**) and spindle-shaped cells in the secretory duct (**d**). **e** The coiled portion of the eccrine gland was characterized by αSMA immunoreactivity (magenta), Acti-stain (green) and Hoechst 33342 staining (blue). High-magnification images of the secretory (**f**) and excretory (**g**) ducts in **e**. αSMA was abundant in the spindle-shaped myoepithelial cells of the secretory duct (**f**, red dashed lines) but absent from the excretory duct (**g**, the excretory duct is outlined with white dotted lines). **h** Acti-stain (middle) and Hoechst 33342 staining (right) distinctly labeled the secretory and excretory ducts with respect to the cell shape and density of nuclei. Arrows indicate the excretory duct, and arrowheads indicate the actin-rich secretory duct.
**i** High-magnification images of the transition region between the SD and ED (boxed in **h**). The red dotted line indicates the border. The arrowheads indicate the elliptical nuclei of myoepithelial cells of the secretory duct, and the arrows indicate the round nuclei of the luminal cells of the excretory duct.

motions from time-lapse images, we adopted the color aberration measurement of two merged images pseudocolored green or magenta at different time points (Fig. 3a). In the control condition with phosphate-buffered saline (PBS), spontaneous motion of the Acti-stained myoepithelial cells was rarely observed, leaving the colors almost completely merged (*white*) for both myoepithelial actin structures and nuclei (Fig. 3b; Supplementary movie 1). Upon pilocarpine stimulation, however, spindle-shaped myoepithelial cells responded and moved slightly, as evidenced by

color aberration from *white* to *green-magenta* (Fig. 3c; Supplementary movie 2). The round nuclei of luminal secretory cells were also dislodged (Fig. 3c). The pilocarpine-induced dislodgement of myoepithelial cells and luminal secretory cells was observed irrespective of the presence or absence of divalent metal ions in the extracellular milieu (Supplementary movie 3). Furthermore, the nuclei of myoepithelial cells without Acti-stain costaining showed greater displacement after pilocarpine treatment (Fig. 3d, e; Supplementary movie 4), suggesting that Acti-stain that directly binds to filamentous actin might interfere with the dynamic motion of actin filaments in myoepithelial cells. Thus, to visualize the skeletal frame of the eccrine gland components, we employed the alternative fluorescent dye CellMask[TM] Deep Red (CMDR), a cell membrane marker[10], which should not interfere with physiological actin dynamics. CMDR staining visualized the αSMA-positive myoepithelial cells of the secretory duct as thick striped streaks (Fig. 4a, b) and the αSMA-negative cuboidal cells of the excretory duct as cubic lattices by delineating the outlines of the cells (Fig. 4c). Therefore, the shape and location of the two duct types were distinctively specified by the CMDR staining patterns. The CMDR-stained eccrine glands contracted more dynamically after pilocarpine treatment than those stained with Acti-stain (Fig. 4d, e; Supplementary movie 5). Notably, contractile motions were most pronounced in the region where the eccrine gland displayed concertina-like CMDR signals (Fig. 4d, *red arrowheads*). Previously, we reported that PGP9.5-positive nerve fibers were predominantly wrapped around the secretory portions of sweat glands[5]. Consistent with this observation, double immunostaining for PGP9.5 and αSMA confirmed that the PGP9.5-positive nerve fibers exclusively surrounded the secretory ducts transversely, while αSMA-positive myoepithelial cells were longitudinally aligned like thick stripes on the secretory duct (Fig. 4f). In addition, the color aberration of Hoechst 33342 signals revealed nuclear displacement in the secretory duct that was triggered by pilocarpine (Fig. 5a, *upper panels*; Supplementary movie 6). Such nuclear displacement was almost completely blocked in the presence of atropine, an antagonist against muscarinic acetylcholine receptors (Fig. 5a, *lower panels*; Fig. 5b; Supplementary movie 7).

Perspiration can also be induced via the activation of nicotinic acetylcholine receptors in vivo[11]. To assess the contribution of nicotinic stimulation to the contraction of the human eccrine gland, we examined the effect of nicotine by 3D live-tissue imaging. The nuclei did not move after nicotine stimulation (Fig. 5c, d; Supplementary movie 8), indicating that muscarinic stimulation, but not nicotinic stimulation, induces the contractile motion of the human eccrine gland as the basis for sweat expulsion[12,13]. Hereafter, we used muscarinic stimulation by pilocarpine to induce eccrine gland contraction in vitro.

**Myoepithelial cells segmentally contract to compress the secretory duct.** At low magnification, the secretory duct was visualized by CMDR staining in two distinctive patterns: a concertina-like pattern with nerve fibers wrapped around the duct in a bandage-like manner[5] (Fig. 6a, *magenta box*) and a thick striped pattern with a faint contour of spindle-shaped myoepithelial cells (Fig. 6a, *yellow box*). In sharp contrast, the excretory duct was visualized in a latticework pattern (Fig. 6a, *black arrowhead*). Upon pilocarpine stimulation, only the concertina-like part of the secretory duct exhibited dynamic contractile motion (Fig. 6b, Supplementary movies 9-11). For quantification of contractile motion, we measured the length of two virtual reference points arbitrarily selected from the bright signals on the secretory duct. The spans of the virtual reference points on the concertina-like part began to shorten after pilocarpine

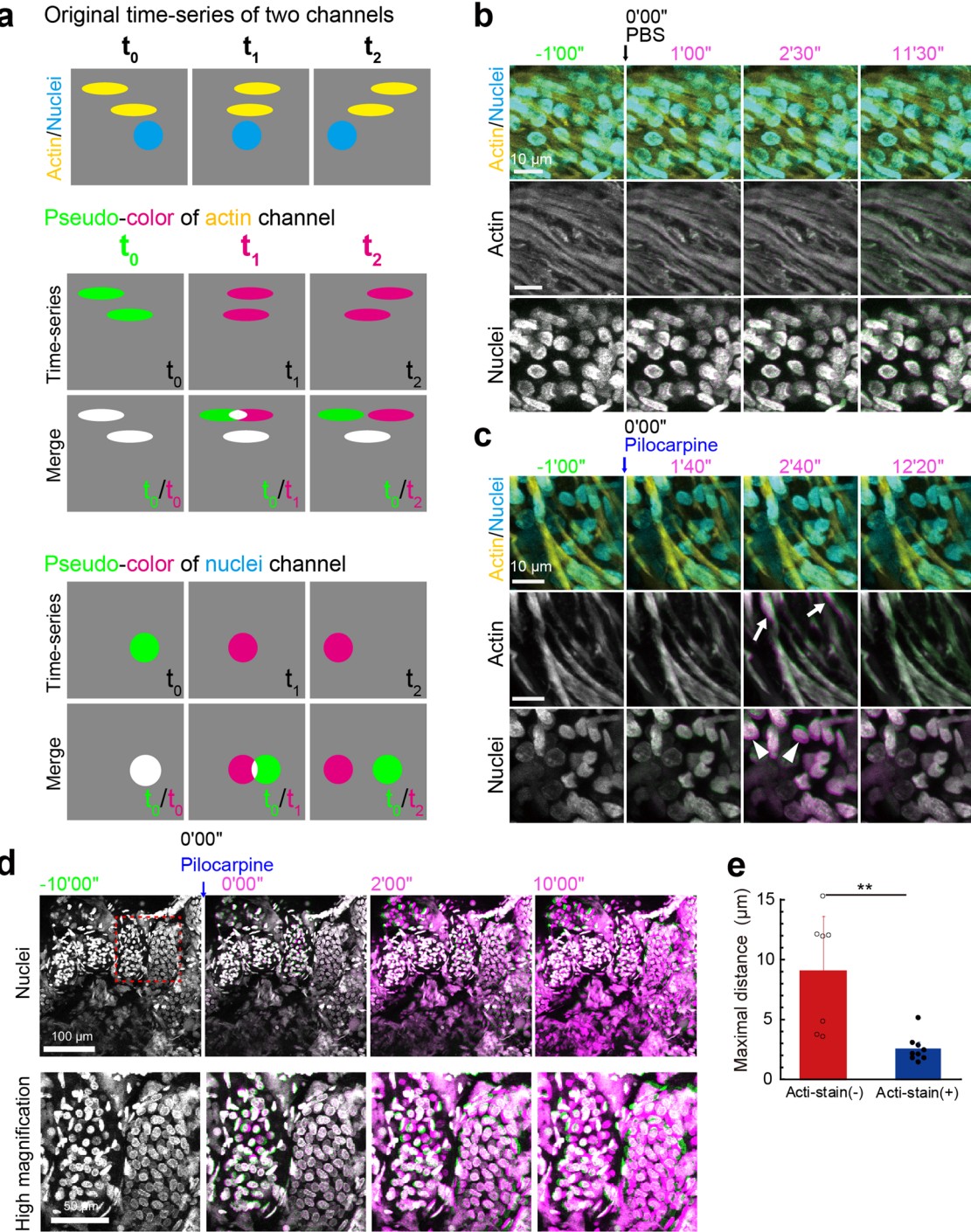

**Fig. 3 Cholinergic stimulation triggers contractile motion of Acti-stained eccrine glands. a** Strategy to detect cell and nuclear displacement by color aberration. The top scheme shows the original video images of two merged channels. The middle scheme shows pseudocolored images of the actin channel at the starting point ($T_0$, green) and at other time points ($T_1$ and $T_2$, magenta). When the images in magenta are merged with the initial image in green, cell displacement can be detected by the appearance of color aberrations. Cells that did not move are observed as entirely white due to the complete overlap of green and magenta. The bottom scheme shows pseudocolored images of the nuclei, showing color aberration. **b** Representative sequential time-lapse images of an eccrine gland after the application of PBS (control). The arrow indicates the time point at which PBS was added ($t = 0'00''$). The upper panels are overlaid fluorescence images of Acti-stain (yellow) and Hoechst 33342 (light blue). The lower panels show the color aberration images for actin (middle) and nuclei (bottom). The merged images remained white, indicating that actin filaments and nuclei did not move. **c** Representative sequential time-lapse images after 10 mM pilocarpine treatment. The panels are aligned in the same order as in **b**. The merged images yielded color aberration, indicative of displacement of actin filaments (arrows) and nuclei (arrowheads) after pilocarpine stimulation. **d** Hoechst 33342-stained eccrine gland. Representative sequential color-aberration images without Acti-stain (red box in the left image). The blue arrow indicates the time point of pilocarpine addition ($t = 0'00''$). **e** The maximum displacement of nuclei with (blue, $n = 9$) or without (red, $n = 7$) costaining with Acti-stain. **\*\***$P < 0.001$, unpaired $t$ test. Panels (**b**), (**c**), and (**d**) have a temporal resolution of 30 seconds.

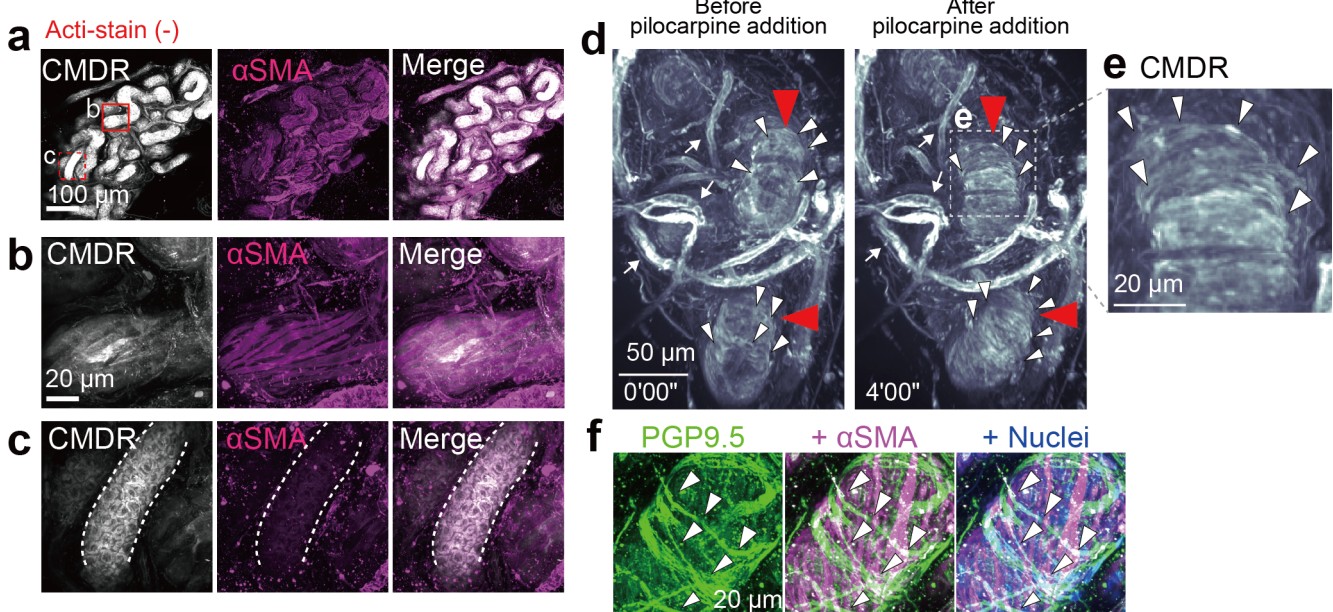

**Fig. 4 Dynamic contraction of the secretory duct occurs at the concertina-like region surrounded by coiled nerve fibers. a** Eccrine gland costained with CMDR and for αSMA. **b**, **c** Magnified views of the secretory duct (**b**; red solid box in **a**) and the excretory duct (**c**; red dashed box in **a**). The white dashed lines indicate the outline of the excretory duct. **d** Time-lapse images of a CMDR-stained eccrine gland before (left) and after (right) the addition of 10 mM pilocarpine. The red arrowheads indicate the position of duct deformation. The white arrows indicate the blood vessels. The white arrowheads indicate concertina-like signals in the secretory duct. **e** Magnified image of the white dotted box in **d**. **f** The secretory duct costained to visualize PGP9.5 (green), αSMA (magenta) and nuclei (blue). White arrowheads indicate PGP9.5-positive signals.

application, reached maximum contraction after approximately 5 minutes, and then gradually returned to the initial length over the next 20 minutes (Fig. 6c). On the other hand, the spans of the virtual reference points on the secretory duct showing thick stripes/less concertina-like patterns remained unchanged (Fig. 6d). Such transient glandular contraction of the concertina-like part was also observed in another pilocarpine-induced contractile motion of the secretory duct (Supplementary movie 5 and Fig. S1). Thus, only the limited parts of the secretory duct characterized by concertina-like patterns engage in pilocarpine-induced contraction.

To further dissect the fine cellular motions within the secretory duct, the displacement of the nuclei of the myoepithelial and luminal cells was traced in magnified cross-sections of the duct. Focusing on the luminal cells, the radius of curvature was estimated from the position of three nuclei (*dots* in Fig. 6e). The radius of curvature of the duct decreased after pilocarpine treatment (Fig. 6f; $P < 0.05$). To further scrutinize the mode of nuclear displacement at the single-cell level, we also compared the direction of nuclear displacement in both myoepithelial and luminal cells. The elliptical myoepithelial nuclei diagonally slid along the perimeter of the duct (Fig. 6g, h). On the other hand, the round luminal nuclei slipped aside after the addition of pilocarpine, followed by a change in direction straight toward the center of the lumen (Fig. 6g, h), suggesting that myoepithelial displacement pushed luminal cells toward the lumen to compress the duct. There was no significant delay in the displacement of the nuclei between myoepithelial cells and their adjacent luminal cells (Fig. 6g, h), which was further supported by the measurement of cumulative distances of the displaced nuclei of myoepithelial cells and their adjacent luminal cells (Fig. 6i).

To substantiate the role of actin dynamics in myoepithelial cells, we pharmacologically inhibited the molecular machinery of smooth muscle contraction. By pretreating the isolated eccrine glands with cytochalasin D, an inhibitor of actin polymerization,

the contraction of the secretory duct and the nuclear displacement of the myoepithelial and luminal cells were abolished, as the color aberration remained white even after pilocarpine stimulation (Fig. 6j, k; Supplementary movie 12). These results indicate that actin dynamics in myoepithelial cells play a pivotal role in the contractile motion of the limited parts of the secretory duct upon cholinergic stimulation.

**Myoepithelial contraction causes reciprocal changes in the lumen volume of secretory and excretory ducts.** To further address the role of the contractile motions of the eccrine gland in sweat secretion, we analyzed the deformation of the CMDR-stained cellular latticework, focusing on the changes in the volume of the luminal space within the duct of the coiled structure. Cell membrane staining by CMDR revealed temporal changes in the latticework of the cellular structure and the lumen volume as signal voids inside the secretory coil. We extracted the areas of the signal voids from the 3D time-lapse data to trace the successive changes in the lumen volume in both the secretory and excretory ducts, thereby capturing the motion of sweat transport driven by the contraction of the secretory duct (Fig. 7a). First, we disassembled the 3D live-tissue images along the axes of time and Z slices; then, the lumen area on each Z slice was extracted by masking the cellular signals and measured on the X-Y plane (Fig. 7b). The lumen volume was estimated from the summation of these X-Y plane areas. The reconstructed 3D image data showed that the lumen of the secretory duct was segmented by partition (Supplementary movie 13; Fig. 7c; *yellow arrowhead* at $t = -8'00''$). On pilocarpine administration, the narrow segment of the lumen started to shrink immediately (Fig. 7c, *arrow* at $t = 0'00''$; *asterisk* (*) at $t = 2'00''$, Fig. 7c, f) and then dilated even larger than the initial volume until the adjacent segments were recanalized (*open triangle* (Δ) at $t = 9'20''$, Fig. 7c, f). The lumen started to shrink again at 9'20''-10'40'' (*open diamond* (◊), Fig. 7c, f) and continued to regress throughout the stimulation. In parallel

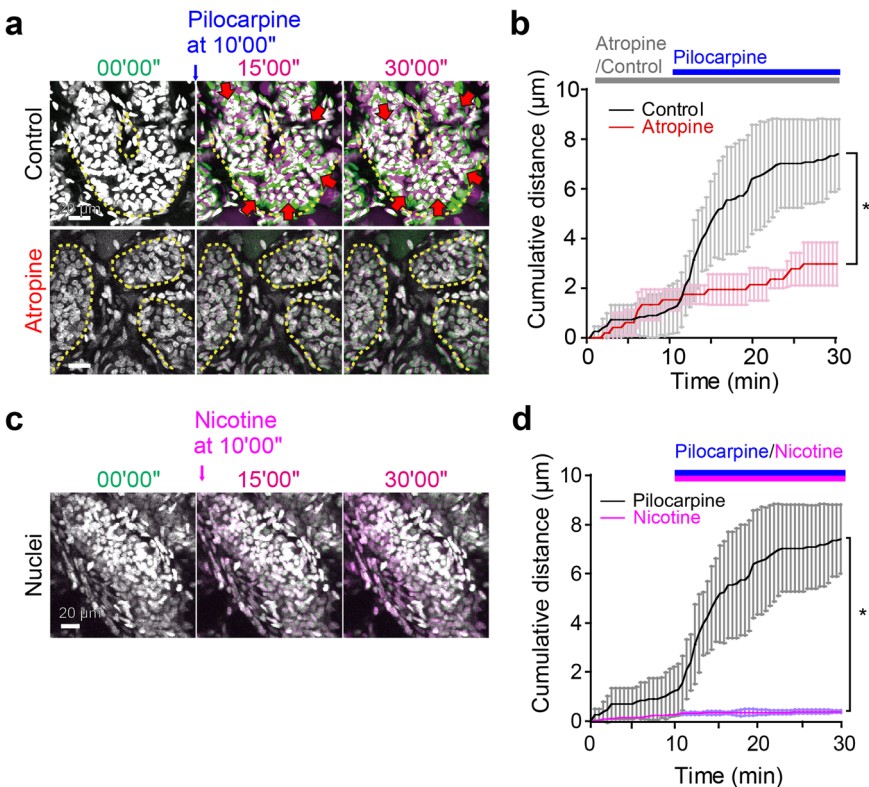

**Fig. 5 Muscarinic stimulation triggers contractile motion of the secretory duct. a** Sequential color aberration images showing pilocarpine-induced displacement of nuclei in the absence (upper panels) or presence (lower panels) of atropine. Pilocarpine was added at $t = 10'00''$ (blue arrow). Red arrows indicate the direction of nuclear displacement. Atropine was added 30 minutes prior to pilocarpine stimulation. **b** Quantification of cumulative nuclear displacement in the control group (black, $n = 5$) and the group pretreated with atropine (red, $n = 5$). The data are shown as the mean ± SD. Pilocarpine was added at $t = 10'00''$. **c** Sequential color aberration images showing the displacement of nuclei in the secretory duct stimulated by nicotine at $10'00''$. **d** Quantification of nuclear displacement in the control group (black, $n = 5$) and the nicotine-treated group (magenta, $n = 5$). Statistics: unpaired $t$ tests; $*P < 0.0002$ (**b**), $*P < 0.000002$ (**d**). Panels (**a**) and (**c**) have a temporal resolution of 30 seconds.

with this second shrinking phase, the different parts of the excretory duct (designated ED1 and ED2) expanded sequentially with a time lag of 4–5 minutes (*closed triangles* (▲), Fig. 7d–f), followed by the gradual shrinking phase that was prolonged over 30 minutes. Thus, pilocarpine-stimulated dilation/constriction of the secretory duct was followed by sequential dilation/constriction of the different parts of the excretory duct, indicative of successive transport of sweat within the coiled secretory portion of the eccrine gland. The dynamics of luminal dilation and constriction support the view that the pilocarpine-induced contraction of myoepithelial cells generates a sweat-transporting force through the excretory duct.

**Gap junction inhibitors block myoepithelial contraction.** Although cholinergic stimulation was a trigger for myoepithelial contraction, not all myoepithelial cells contracted simultaneously. Rather, propagation of cholinergic stimulation on myoepithelial cells may be locally restricted along the longitudinal axis of the secretory duct and induce segmental contractions to generate force for sweat transport. In addition to involuntary control by the autonomic nervous system[14], some smooth muscle cells are electrically interconnected via gap junctions to operate in a coordinated fashion as a functional cluster[15]. Accordingly, excitation can rapidly spread among unitary smooth muscle cells, resulting in segmental contraction. To determine whether the myoepithelial cells of the secretory duct are also interconnected by gap junctions, we examined the effects of carbenoxolone (CBX), a broad-spectrum inhibitor of gap junctions[16], on the contractile motion of the secretory duct upon pilocarpine stimulation. In the presence of CBX, both

luminal and myoepithelial nuclei scarcely moved from their initial positions after pilocarpine treatment (Fig. 8a, b; Supplementary movies 14, 15). The expansion of the excretory ducts was also inhibited in the presence of CBX. The blockade of nuclear displacement by CBX was of a magnitude similar to that imposed by atropine pretreatment (Fig. 8c).

To further explore the role of gap junctions in the segmental contraction of the secretory duct, we examined the expression levels of gap junction proteins, i.e., connexins (Cxs), in the myoepithelial cells of the secretory duct by quantitative reverse transcription polymerase chain reaction (qRT-PCR). Because the myoepithelial cells in the eccrine sweat glands have been characterized as a CD29$^{high}$/CD49f$^{high}$ subpopulation[5], we dissociated the isolated eccrine glands with collagenase and obtained the myoepithelial cell–enriched CD29$^{high}$/CD49f$^{high}$ subpopulation. qRT-PCR analyses of a total of 17 connexins, namely, *Cx26, Cx30, Cx30.3, Cx31, Cx31.1, Cx31.9, Cx32, Cx37, Cx40, Cx40.1, Cx43, Cx45, Cx46, Cx47, Cx50, Cx58,* and *Cx62,* showed that the connexins expressed in the CD29$^{high}$/CD49f$^{high}$ subpopulation were *Cx26, Cx30, Cx31, Cx37, Cx43,* and *Cx45,* of which *Cx43* was most abundant (Fig. 8d). Immunofluorescence staining of the isolated eccrine glands for the connexins expressed in the myoepithelial cells along with αSMA, a marker for myoepithelial cells, showed that Cx43 and Cx45 were detected on αSMA-positive myoepithelial cells as puncta (*white arrowheads*), although no such punctate signals were detected on myoepithelial cells for other connexins, including Cx26, Cx30, Cx31, and Cx37 (Fig. 8e–j). Punctate signals for other connexins were also detected in the luminal cells of the secretory duct at variable

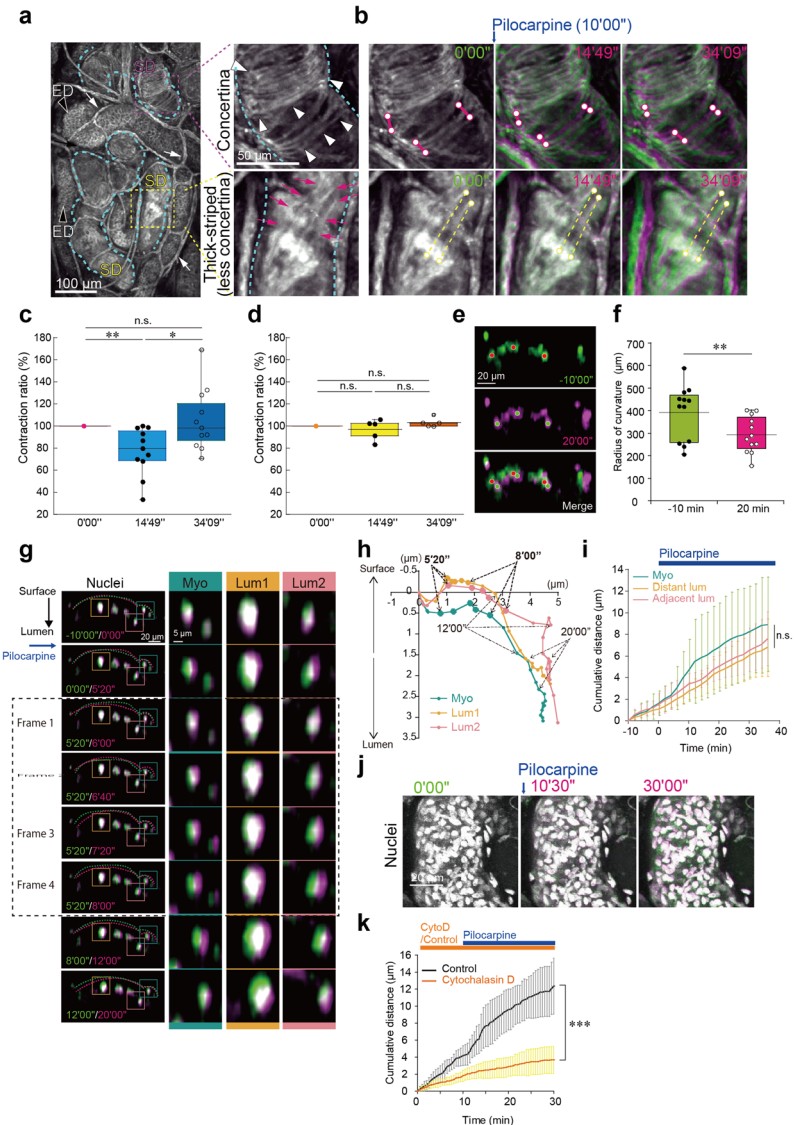

**Fig. 6 Myoepithelial contraction compresses the secretory duct. a** Two distinct patterns of the CMDR-stained secretory duct in the eccrine gland. The secretory duct (SD), contoured with dotted lines in light blue, was stained by CMDR in two distinct patterns: concertina-like (boxed in magenta) and thick-striped/less concertina-like (boxed in yellow). High-magnification images of the boxed area are shown to the right. The excretory duct (ED), visualized in latticework, is indicated by black arrowheads. White arrows (left) indicate blood vessels. White arrowheads (upper right) and red arrows (lower right) point to concertina-like and thick striped/less concertina-like SMDR signals, respectively. **b** Sequential color aberration images of the secretory duct showing concertina-like and thick striped/less concertina-like SMDR signals. Pairs of dots connected by red lines (upper panels) and yellow dotted lines (lower panels) are virtual reference points for quantification of the contractile motion of the secretory duct. Representative color aberration images at 0'00", 14'49", and 34'09" are shown. Quantification of the length between each pair of virtual reference points in the concertina-like (**c**; $n = 11$) and thick stripe/less concertina-like (**d**; $n = 5$) regions of the secretory duct. The contractile motions are expressed as the ratio of the length at each time point to the length at 0'00". **e** Displacement of the nuclei within the same cross-section of the secretory duct before (-10'00") and after (20'00") pilocarpine stimulation. Three nuclei of luminal cells, labeled in red (-10'00") and green (20'00"), are selected for calculation of the radius of curvature of the duct. **f** The radius of curvature of the secretory duct calculated from the coordinates of three selected nuclei of luminal cells before (-10'00") and after (20'00") the addition of pilocarpine ($n = 12$). The dots represent the average values. **g** Stepwise color aberration of the nuclei of luminal and myoepithelial cells on the cross section. Magnified images of the selected myoepithelial (Myo) cells (boxed in green) and adjacent luminal cells (Lum1 and Lum2; boxed in orange and pink, respectively) are shown in the right panels. The start and end points for each time frame are indicated in green and magenta, respectively, for color aberration of nuclei. Boxed in dotted line are consecutive time frames captured every 40". The surfaces of the duct at the start and end time points are outlined with dotted lines in green and magenta. **h** Directions of displacement of the nuclei of myoepithelial cells (green, $n = 6$) and those of two luminal cells, Lum1 (orange, $n = 6$) and Lum2 (pink, $n = 6$), after pilocarpine application. **i** Cumulative distances of displaced myoepithelial cells (green, $n = 6$) and distant (orange, $n = 6$) and adjacent (pink, $n = 6$) luminal cells. j Color aberration images of the nuclei of the secretory duct pretreated with 10 μM cytochalasin D 30 minutes prior to pilocarpine stimulation. **k** Quantification of cumulative nuclear displacement in the control group (black, $n = 13$) and in the group pretreated with cytochalasin D (orange, $n = 14$). **c**, **f**, **k** The data are shown as the mean ± SD. Statistics: unpaired $t$ tests; *$P < 0.01$ (**c**), **$P < 0.05$ (**f**). ***$P < 0.001$ (**k**). n.s. in (**i**), $p = 0.3114$ (Friedman test). The temporal resolution is 40 seconds.

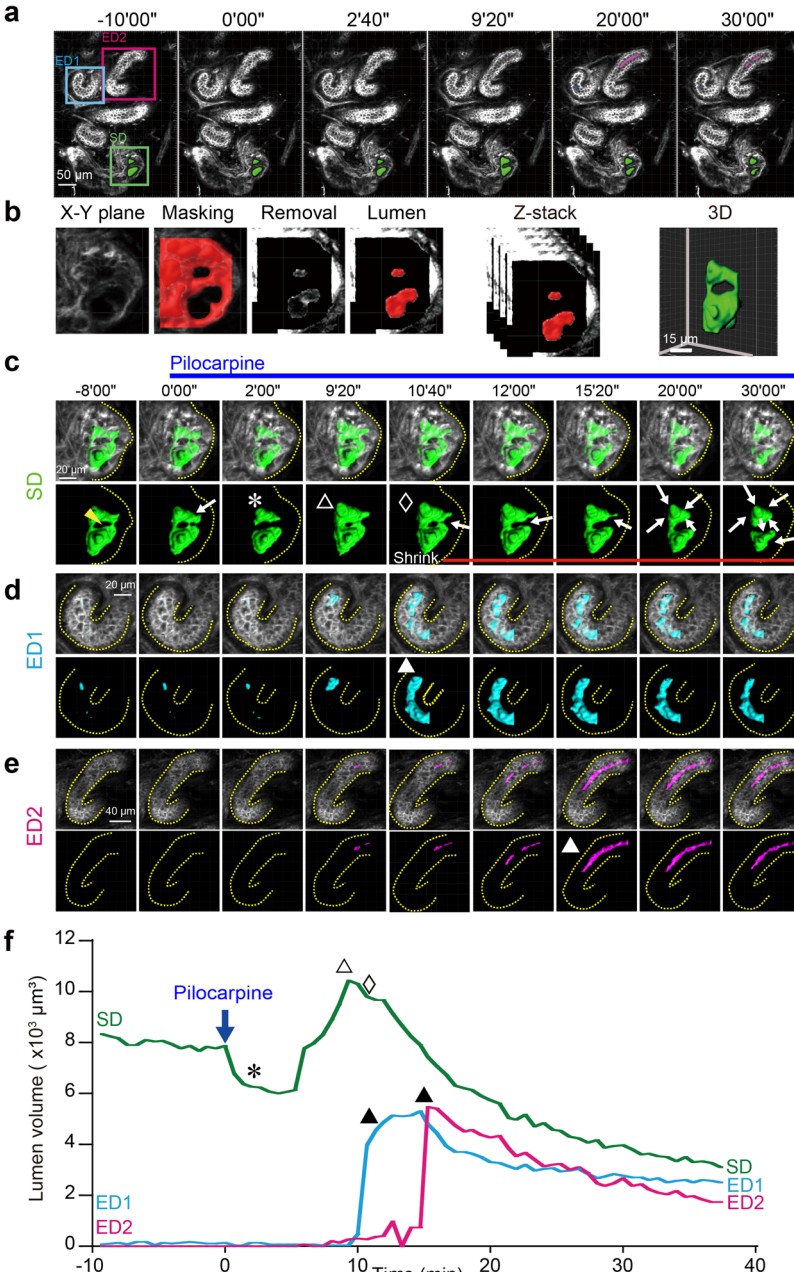

**Fig. 7 Reciprocal changes in lumen volume of secretory and excretory ducts. a** Time series of a representative plane of an eccrine gland stained with CMDR showing the secretory duct (SD), in which the lumen is pseudocolored in green. Different parts of the excretory ducts (ED1 and ED2) are boxed in magenta and blue. **b** Procedural steps to extract the lumen areas as signal voids from an X-Y plane using Imaris software. The time-lapse 3D data in (**a**) were decomposed into multiple planes (X-Y) at different depths. Each plane was processed as follows: First, CMDR signals were rendered ("X-Y plane") and digitally masked ("Masking"), followed by removal of the masked area from the rendered X-Y image ("Removal"). The lumen area was determined as unmasked remnants ("Lumen"). The X-Y panels with the extracted lumen areas were Z-stacked to reconstruct the luminal space for volume measurements ("Z-stack" and "3D"). Time series of perspective views of the Z-reconstructed luminal space in the secretory duct (**c**), ED1 (**d**) and ED2 (**e**) with and without the CMDR-stained cellular framework. The yellow dotted lines indicate the outline of the duct. The white arrows in **c** indicate the direction of compression. **f** Time course of the changes in the lumen volume in the secretory duct (green) and ED1 and ED2 (blue and magenta, respectively). The following symbols denote the sequential contraction/expansion phases in the secretory duct: first contraction (*), expansion (△), and second contraction (◇). ▲ indicates the rapid expansion phase in ED1 and ED2. The temporal resolution is 40 seconds.

densities. Notably, the Cx43 signal was frequently localized between myoepithelial cells in small puncta (Supplementary Fig. S2), raising the possibility that Cx43 is the major gap junction constituent involved in the segmental contraction of myoepithelial cells in the secretory duct.

The role of Cx43 in the segmental contraction of myoepithelial cells was further addressed by using 2-aminoethoxydiphenylborane (2-APB) and Gap27, both of which inhibit Cx43[17,18], although 2-APB has been known to exhibit complex pharmacology, inhibiting not only Cx43 but also inositol trisphosphate receptors (IP3Rs) and transient receptor potential (TRP) channels and having a dose-dependent bimodal effect on store-operated calcium entry[19]. Both inhibitors attenuated the pilocarpine-induced contractile motion of the secretory duct and the nuclear displacement

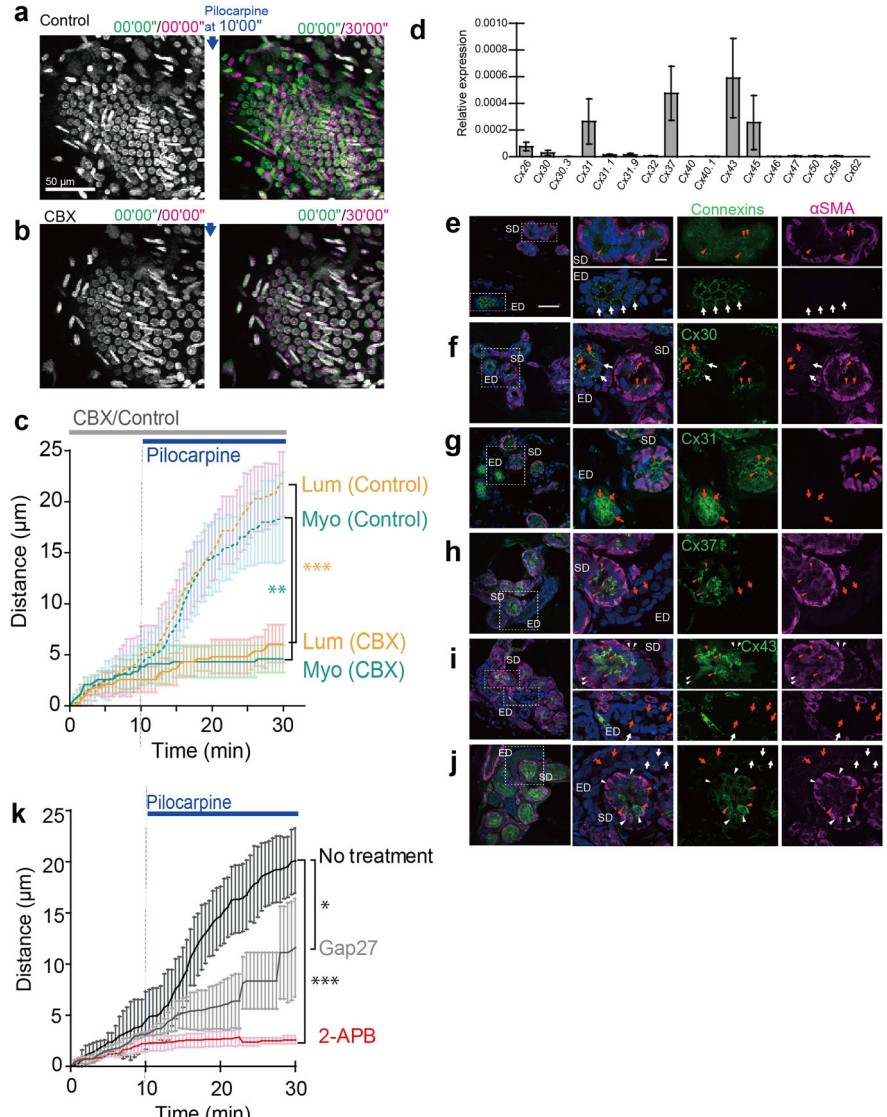

**Fig. 8 Gap junctions mediate pilocarpine-induced contraction of isolated eccrine glands.** Representative color aberration images of pilocarpine-induced contraction of eccrine glands without (**a**) or with (**b**) CBX pretreatment. **c** Quantification of cumulative nuclear displacement in luminal (Lum; orange) and myoepithelial (Myo; blue) cells with (solid line) or without (dotted line) CBX pretreatment ($n = 5$ for each cell type). The data are shown as the mean ± SD. CBX was added at $t = 0'00''$, and pilocarpine was added at $t = 10'00''$. **d** Relative expression of connexin genes in the myoepithelial cell–enriched CD29$^{high}$/CD49f$^{high}$ cell subpopulation ($n = 3$). The expression levels were normalized to 18 S ribosomal RNA. **e–j** Immunofluorescence detection of Cx26, Cx30, Cx31, Cx43, and Cx45. Sections of human eccrine glands were stained for connexins (green) and αSMA (magenta), the latter of which specifically labeled myoepithelial cells of the secretory duct. Nuclei were counterstained with Hoechst 33342. Arrowheads point to connexins detected in puncta on myoepithelial (white) and luminal (red) cells of the secretory duct (SD). White and red arrows indicate connexins on the basal and luminal cells of the excretory duct (ED), respectively. **k** Quantification of cumulative nuclear displacement in luminal cells with and without pretreatment with gap junction inhibitors specific to Cx43, 2-APB and Gap27. $n = 5$ for each treatment condition. The data are shown as the mean ± SD. Pilocarpine was added at $t = 10'00''$. Statistics: unpaired $t$ test; *$P < 0.05$ (**k**), **$P < 0.001$ (**c**), ***$P < 0.0001$ (**c**, **k**).

of the luminal cells (Fig. 8k; Supplementary movies 16, 17), supporting the involvement of Cx43 in the pilocarpine-induced segmental contraction of myoepithelial cells in the secretory duct. The inhibitory effect was more pronounced with 2-APB than with Gap27, an inhibitor more specific for Cx43, suggesting that the complex pharmacological effects of 2-APB on IP3Rs and store-operated calcium entry were also involved in the pronounced inhibition of contractile motion and nuclear displacement in the secretory duct.

Finally, we examined the contribution of gap junctions to sweating in vivo. We applied CBX to the human palm and measured the perspiration rate by the minor test, which visualizes

the expulsion of sweat drops by the metachromasia of the iodine-starch reaction (Fig. 9a). Sweating was induced by autonomic stimulation by passive heating via a foot bath. Prior to the sweating test, we ran a pretest to visualize active sweat pores on the palms of the subjects. Palm sweating was locally inhibited by approximately 30% by pretreatment with 16 mM CBX for 30 minutes compared to the control in a nearby area (Fig. 9b). This result demonstrates that gap junction-mediated communication in eccrine glands is essential for the transportation of sweat to the skin surface. Taken together, our findings suggest that myoepithelial cells are partially connected through gap junctions to segmentally contract for sweat secretion upon autonomic stimulation.

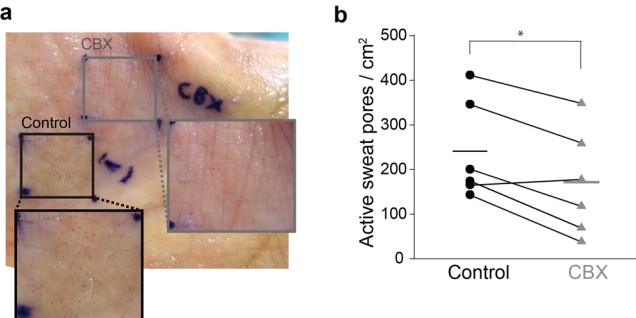

**Fig. 9 CBX suppresses sweat secretion. a** Representative results of the iodine-starch test on human palm. Active sweat pores were visualized as brown dots upon perspiration. Systemic sweating, including palm sweating, was induced by passive heating in a footbath. **b** Effect of CBX pretreatment on palm sweating. The number of active sweat pores in 1 cm² was counted. The bars represent the average. **Statistics**: paired *t* test; *P < 0.01 (n = 6; four males and two females; average age = 31.2 years).

## Discussion

Although the motions of eccrine glands have been investigated by several methodologies[20–22], no study reported to date has captured the live motion of eccrine glands. In the present study, we established a method for ex vivo 3D live imaging to observe the overall contractile motions of isolated human eccrine glands. The dynamic motion of the coiled structure upon cholinergic stimulation was successfully captured, recapitulating the physiological contractile response to autonomic stimuli[23]. The dynamic motion was verifiably the direct consequence of cholinergic stimulation because atropine, a cholinergic antagonist, blocked the motion of the coiled structure. We found that some parts of the secretory duct were visualized on CMDR staining in a concertina-like pattern, which likely reflects sympathetic nerve fibers surrounding the secretory duct. The secretory coils of eccrine glands have been shown to be innervated by sympathetic nerves with varicose nerve terminals[5,24]. The concertina region in the secretory duct displayed more pronounced contractile motion upon cholinergic stimulation than the region with little or no concertina-like signal, consistent with previous ex vivo studies demonstrating glandular contraction[22] and sweat secretion[25] induced by cholinergic stimuli. Our results also revealed that contractile motion of the secretory duct was blocked by gap junction inhibitors, including CBX, 2-APB, and Gap27. The inhibitory effect of CBX was also confirmed by the in vivo perspiration test on human palms. These results raise the possibility that gap junction-mediated communication between myoepithelial cells controls the effective propagation of contraction in a single long tubular structure. Our ex vivo 3D live imaging is therefore a useful model for deciphering the mechanisms that operate during sweat secretion by eccrine glands and the role of myoepithelial cell contraction in transporting sweat to the skin surface.

Several findings indicate that myoepithelial cells surrounding the exocrine secretory ducts and/or alveoli contract upon neuronal or hormonal stimulation[26–29]. Myoepithelial cells are enriched in actin filaments and arranged longitudinally in parallel around the secretory duct of the eccrine gland. We demonstrated that cytochalasin D, an inhibitor of actin polymerization, blocked the pilocarpine-induced contractile motion of the secretory duct of the isolated coiled structure. Acti-stain, a fluorochrome-conjugated phalloidin, also reduced the contractile response to cholinergic stimulation, supporting the role of myoepithelial cells in the contractile motion of the isolated coiled structure. Previously, Sato et al. examined the contractile properties of an isolated secretory duct of the primate palm sweat gland using a

sensitive mechanical strain gauge transducer and demonstrated that acetylcholine, but not adrenergic agonists or the calcium ionophore A23187, induced glandular contraction of the secretory duct[21]. Consistent with this observation, our 3D live imaging detected shortening of the secretory duct in the region showing a concertina-like pattern upon CMDR staining. Such shortening of the secretory duct was not significantly detected in the region that did not show a clear concertina-like pattern. Furthermore, the high-resolution analysis focusing on the displacement of the nuclei of luminal and myoepithelial cells in cross-sections indicated that the curvature radii of the secretory duct decreased after pilocarpine treatment in the following manner: myoepithelial nuclei slid diagonally along the perimeter of the secretory duct, followed by sideways sliding and subsequent downward displacement of luminal nuclei. These observations indicate that contraction of myoepithelial cells causes not only shortening but also narrowing of the secretory duct.

The contraction of myoepithelial cells is considered instrumental in the secretory functions of other exocrine organs, including mammary, salivary, and lacrimal glands[27,28,30,31]. Myoepithelial cells in mammary glands contract in response to the pulsatile release of oxytocin from the pituitary gland to eject milk. Mice deficient in αSMA (ACTA2), the most abundant actin isoform in the mammary duct, exhibited no gross anatomical defects but presented lactation failure[26]. The mammary gland myoepithelial cells from ACTA2-null dams contracted considerably less than those from wild-type dams[26]. Similar lactation defects with reduced contractility of myoepithelial cells have been reported in mice deficient in Orai1, a store-operated Ca²⁺ channel subunit required for milk ejection[32]. It should be noted that isolated sweat glands from Orai1-deficient mice exhibited a comparable calcium release from ER stores upon acetylcholine stimulation with that from wild-type mice[33]. In our study, we observed pilocarpine-induced contractile motion of isolated sweat glands in Ca²⁺-free PBS. Because intracellular calcium was not depleted by EGTA in our experimental conditions, such initial calcium release from ER stores by pilocarpine stimulation may be sufficient for the induction of the contractile motion of the isolated sweat glands. Notably, mammary myoepithelial cells have been shown to communicate with each other through gap junctions[34]. Given that gap junction inhibitors attenuate the contractile motions of the isolated eccrine secretory coil, it is conceivable that the coordinated contraction of myoepithelial cells via gap junctions operates in both mammary and sweat glands to expel secretory products.

The mechanistic basis of how sweat is transferred from the secretory duct to the excretory duct remains to be elucidated. Our 3D image reconstruction analyses focusing on the cholinergic stimulation-induced changes in luminal volume demonstrated that the lumen of the secretory duct was segmented by partition and underwent biphasic contraction with a rebound relaxation period upon cholinergic stimulation, followed by the progressive dilation-regression cycle of the excretory duct. These results raise the possibility that the secretory duct of eccrine glands is divided into segments and contracts segmentally upon cholinergic stimulation to transport sweat to the excretory duct.

Consistent with the scheme, the nerve fibers wrap around the secretory coil in a discrete manner to yield CMDR staining patterns with and without (or less) concertina-like signals, thereby eliciting the segmental contraction of the highly entangled secretory coiled structure. Such segmental contraction may effectively summate the compression force of the segmented portion to transport the secreted sweat toward the excretory duct.

Previous in vivo studies demonstrated high-frequency pulsatile sweat expulsion in humans[35,36], which has been shown to occur

in response to the firing rhythm of skin sympathetic nerve activity[37]. Such pulsatile sweat expulsion can be elicited either by rhythmic contraction of myoepithelial cells or by sweat production by secretory cells in a pulsatile rhythm. Segmental contraction of the secretory duct, together with intermittent cholinergic stimulation, is instrumental to ensure high-frequency pulsatile sweat expulsion from the eccrine gland.

Perspiration can be induced not only by agonists of muscarinic acetylcholine receptors but also by an agonist of nicotinic acetylcholine receptors, i.e., nicotine[11]. Both muscarinic and nicotinic acetylcholine receptors were detected in eccrine sweat glands by in situ hybridization and immunohistochemistry[12]. Our results show that nicotinic stimulation does not trigger the contractile motion of the secretory duct, making it likely that nicotinic stimulation governs sweat production from secretory cells but not the contraction of myoepithelial cells. Consistent with this conclusion, Sato and Sato (1981) reported that isolated eccrine sweat glands contracted in response to muscarinic agonists (e.g., muscarine and pilocarpine) but not in response to nicotine[25]. They concluded that sweating in response to nicotine was elicited through the axon reflex, as proposed by Coon and Rothman[11].

Our results show that the segmental contraction of myoepithelial cells requires gap junctional intercellular communication (GJIC), as CBX, 2-APB, and Gap27 inhibited the contractile motions of the isolated secretory coil upon cholinergic stimulation. qRT–PCR analyses of the connexins expressed in the myoepithelial cell–enriched CD29[high]/CD49f[high] subpopulation indicate that the myoepithelial cells express Cx26, Cx30, Cx31, Cx37, Cx43, and Cx45, of which Cx43 is most abundant. Immunofluorescence staining of Cx43 in the secretory coil of the eccrine gland shows that Cx43 is expressed on the myoepithelial cells in small puncta that are frequently detected between the myoepithelial cells, making it likely that the gap junctions connecting the myoepithelial cells are predominantly composed of Cx43. In support of this possibility, Gap27, a peptide gap junction inhibitor derived from Cx43, blocked the pilocarpine-induced contractile motion of the secretory coil of the isolated eccrine gland. Consistent with this view, there is accumulating evidence that Cx43 is the major connexin type expressed in myoepithelial cells of various exocrine glands[34,38–40]. In mammary glands, while Cx26, Cx30 and Cx32 are expressed in luminal cells, Cx43 is primarily expressed in myoepithelial cells[41] and is involved in myoepithelial contraction for milk ejection[34,40,42]. Thus, ectopic expression of Cx26 suppresses the expression of Cx43 in the myoepithelial cells of mammary glands, leading to impaired GJIC and defects in milk ejection[34]. A similar defect in milk ejection has been reported in a mouse model harboring a missense mutation in the Cx43 gene, which resulted in reduced Cx43 expression in myoepithelial cells[40]. Cx43 has also been observed at gap junctions connecting myoepithelial cells in sublingual glands and implicated in the contraction of myoepithelial cells[39]. Given that Cx43 is also predominantly expressed in the smooth muscle cells of the aorta[43] and is involved in vasoconstriction by vasopressin[44], it is conceivable that GJIC through Cx43 underlies the effective propagation of contractile signals within a segmented cluster of myoepithelial cells in the secretory duct of eccrine glands.

The requirement of GJIC in sweat secretion in vivo was also demonstrated by the perspiration test on a human palm, as topical administration of CBX suppressed sweat secretion by 30%. From a clinical viewpoint, selective inhibitors of Cx43 may be promising as antiperspirants to halt sweating without the drawbacks of conventional treatments. In everyday life, abnormally excessive sweating causes various social problems, including mental stress caused by self-hatred and unnoticed harassment by others in addition to daily inconvenience[45]. However, the sole antiperspirants available for the last century have been conventional metal colloid–type antiperspirants that mechanically plug sweat pores[46]. These mechano-plug antiperspirants are convenient, but they are clinically and esthetically disadvantageous because they irritate the skin and stain clothes[47,48]. Additionally, the sweat held in the ducts can pathologically leak into the dermis, causing acute exudative inflammation[49–51]. In contrast, selective gap junction inhibitors in the form of antedrugs could be safe antiperspirants with minimal adverse effects on the skin. CBX is such a candidate ingredient for next-generation antiperspirants. Notably, chemotherapeutic drugs acting on the hypothalamic–pituitary–adrenal (HPA) axis are known to have an impact on perspiration. For example, many antidepressants that act on the HPA axis have a side effect of causing excessive sweating[52]. Antiepileptic drugs sometimes cause sweating disorders by acting on the hypothalamic–pituitary–somatotropic axis, resulting in diminished sweating and a risk of heatstroke[53,54]. Conversely, it is known that anticholinergic drugs cause diminished sweating and have been used to treat hyperhidrosis, although these drugs cause side effects such as dry mouth, dry eyes and constipation[55]. CBX can be applied as a topical therapeutic agent that acts on the sweat glands directly by blocking GJIC, thereby eliminating the side effects of anticholinergic drugs.

In conclusion, 3D live-tissue imaging of the whole eccrine coiled structure revealed the gap junction-mediated contraction of myoepithelial cells in the secretory duct and progressive dilation-regression cycle of the excretory duct for sweat expulsion. The contraction sequence is thus more complicated than simply compressing the secretory duct, requiring further investigation of the segmental contraction that occurs in the secretory duct.

## Methods

**Human skin specimens**. Fresh human postoperative skin specimens were obtained with written informed consent from all subjects at Osaka University Medical School (Osaka, Japan), Kinugasa Clinic (Osaka, Japan), Seishin Plastic and Aesthetic Surgery Clinic Co., Ltd (Tokyo, Japan) and Bizcom Japan (Tokyo, Japan). Experiments using human skin samples were approved by the Ethics Committee of Osaka University (yakukumi-28-3, yakukumi-28-5, yakukumi-29-2, yakujin-2019-10, yakujin-2019-28). Dissected skin specimens were stored at 4 °C in saline for up to 72 hours before use.

**Eccrine gland isolation**. Human eccrine glands were isolated from fresh skin specimens by microsurgery as follows. Eccrine glands in the dermis were labeled using 10 mM neutral red (Sigma–Aldrich, Gillingham, UK) in divalent cation-free PBS (PBS(-)) for 5–10 minutes prior to microsurgery[56]. The labeled eccrine glands were gently picked up manually from the skin by pinching the surrounding tissues with forceps to minimize damage to the ducts under a stereomicroscope (M-125; Leica, Wetzlar, Germany) and placed in PBS(-) until use in subsequent experiments.

**Staining and mounting live eccrine glands**. Freshly isolated eccrine glands were incubated for 30 minutes at 25 °C in vital stain solution supplemented with either Acti-stain 488 (Alexa 488-conjugated phalloidin; Cytoskeleton, Inc., Denver, USA), CMDR (a cell membrane stain; Thermo Fisher Scientific, Massachusetts, USA) or Hoechst 33342 and then washed with PBS (-) for 10 minutes. The stained eccrine glands were mounted on glass-bottom dishes (μ-Slide 8-well Glass Bottom, #80826, ibidi, Martinsried, Germany) coated with collagen gel (type 1-A: Nitta

Gelatin, Inc., Osaka, Japan), which stably held the isolated eccrine glands. To prevent the samples from drying, the mounted eccrine glands were immersed in Krebs–Ringer bicarbonate solution (KRS) supplemented with 11 mM glucose at 25 °C[18], except that the mounted eccrine glands were immersed in $Ca^{2+}/Mg^{2+}$-free PBS in the experiments shown in Fig. 3 and Supplementary movies 1, 2, and 4.

**Live imaging**. For the time-lapse 3D visualization of eccrine gland dynamics, multicolor fluorescence images of labeled nuclei (excitation: 405 nm), cell membranes (excitation: 650 nm) and actin filaments (excitation: 488 nm) were recorded under a confocal microscope (IX83 with FV1200; Olympus Corporation, Tokyo, Japan) with a 20× objective lens (NA of 0.75; Olympus Corporation) at intervals of 30 or 40 seconds for 30 minutes, depending on the thickness of the samples. At each recording time, twelve to thirty Z slices at 2-μm steps were obtained and immediately stacked to reconstruct the 3D images of the ducts. The Z-stacked images were freely rotated, and any plane of interest was obtained as a single image using ImageJ (NIH, USA). To induce eccrine gland contraction, 10 mM pilocarpine (P6503, Sigma–Aldrich, Missouri, USA) was added 3 or 10 minutes after the initiation of image recording, thereby confirming the absence of any significant spontaneous contraction with the gland tested. To examine the nicotinic effect on sweat gland contraction, 6 mM nicotine (140-01211, Wako, Japan) was added at 10 minutes to start the observation. To inhibit pilocarpine-induced contraction, the eccrine glands were treated with 10 μM atropine (A0257, Sigma–Aldrich), 1 mM CBX (C4790, Sigma–Aldrich), 50 μM 2-APB (013-24911, Wako), or 50 μM Gap27 (HY-P0139, MedChemExpress, USA) 30 minutes before stimulation with pilocarpine. Eccrine gland dynamics were calculated by the object-tracking function of ImageJ, focusing on nuclei. Short-interval time-lapse images of eccrine gland dynamics were captured by confocal microscopy with a spinning disk system (IX83 with CSU W-1; Olympus Corporation and Yokogawa, Tokyo, Japan) and a 20× or 40× objective lens (NA of 0.75 and 0.95, respectively).

**Pharmacological inhibition of myoepithelial contraction**. To confirm whether myoepithelial contraction induces gland dynamics, actin dynamics were inhibited by cytochalasin D (Cayman Chemical, USA), which was dissolved in dimethyl sulfoxide (DMSO, Thermo Fisher Science, USA) and used at 10 μM, where the DMSO concentration in the chamber was 0.1% after dilution with KRS. The mounted eccrine gland was pretreated with cytochalasin D 30 minutes before stimulation with pilocarpine. As the control, 0.1% DMSO was applied for pretreatment at the same time point.

**Calculation of the luminal space from 3D cellular latticework signals**. Original confocal images before Z-stacking were processed to detect the cellular latticework signals as follows. First, the CMDR cell membrane signals were selected and rendered into a continuous surface by Imaris software's Surface tools (Oxford Instruments, Belfast, UK). Within the rendered surface, the areas with continuous signal intensities corresponding to the cell-rich walls of the duct were digitally masked by the Imaris Masking tool. Next, the masked areas were cut out from the images. Then, the remaining cell-free luminal areas were selected and integrated along the depth to calculate the lumen volume of the inner duct in each 3D view along the entire length of the time-lapse movie. The nuclear signals provided information regarding the position of individual cells in motion. The 3D luminal views were imposed on the original 3D data, and any planes of interest were obtained by the Oblique-slicer tool in Imaris. The displacement of the

nuclei was analyzed by the distance-calculation function in Imaris.

**Immunostaining and fluorescent labeling**. Isolated eccrine glands with or without CMDR staining were fixed in 4% paraformaldehyde (PFA) in PBS (Wako Pure Chemical, Osaka, Japan) at 25 °C for 4 hours, washed three times with PBS supplemented with 0.5% Triton X-100 (PBS-T), and blocked with 5% bovine serum albumin (BSA: A7030-100G, Sigma–Aldrich) in PBS-T at 4 °C for 12 hours. Subsequently, the eccrine glands were incubated with anti-αSMA antibody (1:200, ab5694, Abcam, MA) and anti-PGP9.5 antibody (1:200, ab8189, Abcam, MA) in the same blocking solution at 4 °C for 12 hours. After washing with PBS-T three times, the eccrine glands were incubated with Alexa594-conjugated anti-rabbit antibody (1:400, A11037, Thermo Fisher Scientific), Alexa 488-conjugated anti-mouse antibody (1:400, A28175, Thermo Fisher Scientific) and Hoechst 33342 (H3570, Life Technologies, USA) at 25 °C for 12 hours.

**Immunostaining of frozen tissue sections**. Skin tissues were fixed in 4% PFA in PBS at 4 °C overnight, embedded in optimal cutting temperature compound (4583, Sakura Finetek Japan Co., Japan) and frozen in liquid nitrogen–chilled 2-methylbutane. The samples were sectioned at 10 μm using a cryostat (CM3050S, Leica Microsystems, Wetzlar, Germany), fixed in 4% PFA in PBS and blocked with 5% BSA in PBS-T at 25 °C for 20 minutes. For Cx31 staining, frozen skin tissue blocks were used; after sectioning, they were fixed with ice-cold acetone for 15 minutes. Subsequently, the sections were incubated with primary antibodies in the same blocking solution at 4 °C for 12 hours. The antibodies used were anti-Cx26 (1:500, ab59020, Abcam, Cambridge, UK), anti-Cx30 (1:100, HPA014846, Sigma–Aldrich), anti-Cx31 (1:50, LS-C375784, LSBio, WA, USA), anti-Cx37 (1:125, #40-4300, Invitrogen), anti-Cx43 (1:1000, ab11370, Abcam), anti-Cx45 (1:100, CX45B12-A, Alpha Diagnostics) and anti-αSMA (1:100, A2547, Merck). After being washed with PBS-T three times, the sections were incubated at room temperature for 2 hours with the following secondary antibodies and Hoechst 33342 (1:1000): Alexa Fluor 488-conjugated goat anti-mouse IgG and Alexa Fluor 594-conjugated donkey anti-rabbit or anti-goat IgG (1:200, A21202, A21207, A11058, Thermo Fisher). After a final wash, the slides were mounted with Prolong Gold (P10144, Thermo Fisher).

**Measurement of sweating**. The activity of human eccrine glands was measured on the palm of six healthy volunteers (four males and two females) under passive heating by a foot bath with approval from the Ethics Committees of Mandom Corporation (Osaka, Japan). Prior to the heat stress test, active sweat pores were stained by the starch-iodine method (Minor test). Absorbent cotton soaked in 16 mM CBX in 20% ethanol or in 20% ethanol was placed on the palm and held in position for 30 minutes at 25 °C. Then, the absorbent cotton was removed, and the surface of the palm was dried. The palms were stained with 3% w/v iodine (092-05422, Wako Pure Chemical) in ethanol and 10% starch solution in castor oil (034-01586, Wako Pure Chemical). To induce sweating, the volunteers immersed their feet in hot water (42 °C) for 30 minutes[57]. All ethical regulations relevant to human research participants were followed. Signed informed consent was obtained from all participants.

**Statistical analysis**. The statistical analyses were performed using either Microsoft Excel (Microsoft, WA, USA) or KaleidaGraph 4.5 (Synergy Software, PA, USA). All data are presented as the mean ± standard deviation (SD).

**Measuring the radius of curvature**. The coordinates of three continuous nuclei of luminal cells were chosen and plotted on an arbitrary Cartesian plane. These coordinates were applied to obtain the radius of curvature using the online calculation platform Wolfram Alfa (https://www.wolframalpha.com/widgets/view.jsp?id=b0d28dd78e48b231c995d69b91666803).

**Flow cytometry for isolation of CD29high/CD49fhigh myoepithelial cells**. Fresh skin tissues were minced and enzymatically disaggregated for 16 hours at 37 °C in Complete MammoCult Human medium (Stem Cell Technologies, Vancouver, BC, Canada) with 600 U/ml collagenase type II (Worthington Biochemicals, Freehold, NJ, USA) on a tube rotator. After centrifugation at 300× g for 5 minutes, a single-cell suspension was obtained by repeated pipetting for 3 minutes in prewarmed 0.5% trypsin/1 mM EDTA in PBS, followed by repeated pipetting for 1 minute in prewarmed 5 mg/ml dispase (Gibco, Paisley, UK)/0.1 mg/ml DNase I (Sigma). The resulting suspension was filtered through a 40-μm mesh (BD Biosciences, San Jose, CA, USA). Immunolabeling of cells was performed at 4 °C for 30 minutes in PBS containing 5% BSA with allophycocyanin-conjugated anti-CD29 (1:6, 559883, BD Pharmingen, San Diego, CA) and Brilliant Violet 421-conjugated anti-CD49f (1:20, 313624, BioLegend, San Diego, CA). After being washed twice with 5% BSA in PBS, the cells were resuspended in PBS with 7-AAD (1:100, BioLegend) for dead cell staining.

Flow cytometric analysis and cell sorting were performed with a FACSAria2 (BD Biosciences), and the CD29high/CD49fhigh population was sorted for further qRT-PCR analysis.

**Quantitative RT-PCR (qRT-PCR)**. Total RNA was isolated using TRI Reagent LS (Molecular Research Center, Inc.). cDNA was synthesized using the QuantiTect Reverse Transcription Kit (Qiagen, Hilden, Germany). Real-time quantitative PCR was performed using THUNDERBIRD SYBR qPCR Mix (Toyobo, Osaka, Japan) and a ViiA7 real-time PCR system (Applied Biosystems, CA, USA). The expression levels of target genes were normalized to that of 18 S ribosomal RNA using a standard curve method. The primer sets for qRT-PCR are listed in Table 1.

**Reporting summary**. Further information on research design is available in the Nature Portfolio Reporting Summary linked to this article.

## Data availability

Numerical source data for all plots and graphs can be found in the supplementary data file.

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

## Acknowledgements

This work was supported by JSPS KAKENHI Grants-in Aid for Young Scientists (B) (grant number JP17K16337), Early-Career Scientists (grant number JP19K17771) and Grant-in-Aid for Scientific Research (C) (grant number JP22K08429) to K.N. and by the Collaborative Research Fund from Mandom Corporation to K.S. This work was also partly supported by MEXT KAKENHI for Transformative Research Area (A) (23721401) to K.S. We are grateful to Moe Nishizawa for assistance in tissue immunostaining and data acquisition. We thank American Journal Experts (https://www.aje.com/go/springernature) for providing professional manuscript editing.

## Author contributions

K.N. designed and performed ex vivo and in vivo experiments, analyzed the data and wrote the manuscript. H.K., L.Q. and T.H. performed FACS sorting of myoepithelial cells, qRT–PCR analyses, and tissue immunostaining and analyzed the data. R.K. conceived the project, designed the in vivo perspiration test and analyzed the data. F.F. analyzed the data and wrote the manuscript. F.O. analyzed the data and financially supported the project. Y.N., A.T., H.M., and I.K. contributed to the sample preparation. K.S. conceived the project, analyzed the data, and wrote the manuscript. All authors contributed to the preparation of the manuscript.

## Competing interests

The authors declare no competing interests.
