## [Peer review file · Communications Biology]

Reviewers' comments:

Reviewer #1 (Remarks to the Author):

This is a nicely written article with strong evidence on the physiology of nerve-sweat gland activity. Although this is a basic biology journal, please comment on the following:

- There are several chemotherapeutic drugs that affect the HPA axis as well as the peripheral nervous system, does this lead to heat retention in patients?
- What would be the consequence of a treatment to disrupt Cx43

These comments should be discussed

Reviewer #2 (Remarks to the Author):

The manuscript by Nakashima et al. explores contractility of human eccrine sweat gland myoepithelial cells in ex vivo preparations in response to cholinergic stimulation. Overall, the manuscript is well written and presented. Whilst the concepts are not new, visualization in this manner has provided some technological and conceptual advances to this specific field. However, the authors base many of the conclusions on cell identification using non-specific dyes and quantification of cell-type/region specific movement on the background of substantial tissue-level movement with rather crude methods of quantification.

I believe the manuscript could be suitable for publication in this journal, provided the authors address the following comments and soften some of their conclusions.

-Line 118. "In the control condition with phosphate-buffered saline (PBS)"

--What is the concentration of extracellular calcium in these experiments (both control and agonist conditions)? In the discussion the authors acknowledge a potential role for calcium (which makes sense with this agonist and with actin-mediated contraction generally and in light of other work). It is therefore important that physiological levels of extracellular calcium are present.

-Lines 125-128. "Furthermore, the nuclei of myoepithelial cells without Acti-stain containing showed greater displacement after pilocarpine treatment (Fig. 3d, e; Supplementary movie 3), suggesting that Acti-stain that directly binds to filamentous actin might interfere with the dynamic motion of actin filaments in myoepithelial cells."

--Movie S2 looks like obvious contraction. Movie S3 actually looks like the sample has slipped with agonist addition. If the authors are going to make this claim about the stain (influencing future use by others), it should be supported by stronger evidence.

-Lines 137-148. "Notably, contractile motions were more pronounced in the region where the eccrine gland displayed concertina-like CMDR signals (Fig. 4d, red arrowheads). Because nerve fibers have been shown to be wrapped around a part of the secretory coil, we reasoned that concertina-like signals might arise from the sympathetic nerve fibers being wrapped around secretory ducts but not from myoepithelial cells per se. To corroborate this possibility, we costained the secretory duct for α SMA and PGP9.5, the latter of which is a marker of nerve fibers. PGP9.5-positive nerve fibers

exclusively surrounded the secretory ducts transversely, while α SMA- positive myoepithelial cells were longitudinally aligned like thick stripes on the secretory duct (Fig. 4f). These results are consistent with the possibility that concertina-like CMDR signals stem from nerve fibers and that myoepithelial cells are directly governed by cholinergic nerve terminals.”

--This seems obvious. What hypothesis is being disproved here? The wording of this is also a bit strange considering they have used a bath solution.

Line 153-160. “Perspiration can also be induced via the activation of nicotinic acetylcholine receptors in vivo. To assess the contribution of nicotinic stimulation to the contraction of the human eccrine gland, we examined the effect of nicotine by our 3D live-tissue imaging. The nuclei did not move after nicotine stimulation (Fig. 5c, d; Supplementary movie 7), indicating that muscarinic stimulation, but not nicotinic stimulation, induces the contractile motion of the human eccrine gland as the basis for sweat expulsion. Hereafter, we used muscarinic stimulation by pilocarpine to induce eccrine gland contraction in vitro.”

--The authors explore this but do not discuss further. If nicotinic stimulation causes contraction in vivo, but this cannot be observed ex vivo, what is the difference? Is it a technical issue? This should be discussed.

-Paragraph line 162-178.

--I find this entire section unconvincing. Firstly, we are distinguishing only slight differences in the staining patterns of a non-specific stain on live tissue. Secondly, the entire structure moves, making it difficult to see where the contraction is initiated and where movement is active vs passive. The mechanism of quantification cannot distinguish. This is also the case for the next section

-Line 188 “It should be noted that the displacement of the myoepithelial nuclei preceded that of the luminal nuclei”

--what is the temporal resolution for these experiments? Can the authors comment on whether they have the temporal resolution to quantify difference in the movement of one cell type and a cell type that is physically connected and very close to it.

-Line 197. The dominance of green signals in the frame at 30:00 was due to fluorescence quenching after prolonged recording

--Freely available imaging software (e.g., ImageJ) has bleach corrections tools that may be useful. Have the authors inverted the colours between the movie and the figure. It appears like the magenta is dominating in the movie.

-Fig 8

--The IHC is very hard to make sense of. Is this actin staining or alpha smooth muscle actin staining. This would be much more clear with cell type specific stains, particularly in 2D images. For fixed tissue with IHC there is no reason not to use cell-type specific antibodies. This is essential to support their conclusions about cell type specific expression of the various connexin proteins.

Fig. 8h

Can the authors confirm that the regions of human skin used were alternated and this isn't a region specific phenomenon?

Discussion

Lines 297-298

Reference these papers also showing myoepithelial contraction in 3D, which are highly relevant to this study.

Lacrimal gland - <https://doi.org/10.1038/s41598-018-28227-x>

Mammary gland - <https://doi.org/10.1073/pnas.2016905117>

Reviewer #3 (Remarks to the Author):

Nakashima et al., provided evidence for the contractile function of myoepithelial cells in peristaltic sweat secretion from human eccrine glands. By using ex vivo live imaging, the authors provided solid data. But, there are at least two parts of defects (as follows) which prevent the consideration in its current form.

1. The authors did great work in the myoepithelial contraction - induced the peristaltic transport of sweat. However, the authors did not provided enough evidence to show the role of gap junction in this mediation work. The manuscript title emphasized the gap junction mediation which cannot be convincingly supported by the data provided.

2. From the data provided by the authors Figure 8, we can see the blockage of gap junction impact myoepithelial contraction, which indicated the potetial role of gap junction on myoepithelial contraction. Why to chose gap junctions? How about the tight junctions and anchoring junctions?

3. Gap junctions are mainly controled by the profile of connexin family. But the authors did not correctly show the expression differences of Connexin family. Why to choose Cx26, Cx30, Cx32 and Cx43 in myoepithelial cell? How many members of connexin family are expressed in myoepithelial cell? Which member has the highest expression in myoepithelial cell and which member plays the most important role? From Figure 8d-g, we can not judge these information.

4. Besides, there is no other data to explain the underlying regulation mechanism, which is the big defect of the manuscript.

Others

1. Page 15, line 422, "Acti-stain" Writing errors.

Reviewer #1:

*This is a nicely written article with strong evidence on the physiology of nerve-sweat gland activity.*

*Although this is a basic biology journal, please comment on the following:*

*- There are several chemotherapeutic drugs that affect the HPA axis as well as the peripheral nervous*
*system, does this lead to heat retention in patients?*

We thank the reviewer for raising this question. Because perspiration is controlled by the
hypothalamus and spinal cord, chemotherapeutic drugs that affect the HPA axis as well as those that
affect the peripheral nervous system have an impact on perspiration. For example, many
antidepressants that act on the HPA axis cause excessive sweating as a side effect (Beyer et al., 2017
[ref. 49]). Antiepileptic drugs sometimes cause sweating disorders by acting on the hypothalamic–
pituitary–somatotrophic axis, resulting in diminished sweating and a risk of heatstroke (Cerminara et
al., 2006 [ref. 50], Canel et al., 2016 [ref. 51]). Conversely, anticholinergic drugs are known to reduce
sweating and have been used to treat hyperhidrosis, although these drugs cause side effects such as
dry mouth, dry eyes, and constipation (Glaser et al., 2015 [ref. 52]). These side effects of
chemotherapeutic drugs have been discussed in the revised manuscript (page 14, line 432–page 15,
line 439).

It should be noted that chemotherapeutic drugs acting on the hypothalamic–pituitary–
adrenal (HPA) axis are known to have an impact on perspiration. For example, many
antidepressants that act on the HPA axis have a side effect of causing excessive sweating⁴⁹.
Antiepileptic drugs sometimes cause sweating disorders by acting on the hypothalamic–

┌

┐

15

pituitary–somatotrophic axis, resulting in diminished sweating and a risk of heatstroke⁵⁰,
⁵¹. Conversely, it is known that anticholinergic drugs cause diminished sweating and have
been used to treat hyperhidrosis, although these drugs cause side effects such as dry mouth,
dry eyes, and constipation⁵². CBX can be applied as a topical therapeutic agent that acts

┌

┐

*The rising body temperature is detected by the hypothalamus via the blood flow.*

- What would be the consequence of a treatment to disrupt Cx43

We wish to thank the reviewer for this comment. We examined the effects of two Cx43-specific
blockers, 2-aminoethoxydiphenylborane (2-APB) and Gap27, on the contractile motion of the sweat
gland upon cholinergic stimulation. In the presence of 2-APB, nuclear displacement induced by
pilocarpine treatment was blocked in the sweat gland (new **Fig. 8k** and **Supplementary movie 16**).
Furthermore, a similar inhibitory effect, albeit less pronounced, was observed with Gap27 (new **Fig.**
**8k** and **Supplementary movie 17**). These new results are consistent with the expression patterns of
connexins in sweat glands. qRT-PCR analyses showed that the connexins expressed in myoepithelial
cells of the sweat gland were Cx26, Cx30, Cx31, Cx37, Cx43, and Cx45, of which Cx43 was most
abundant (new **Fig. 8d**). Immunohistochemical staining for Cx43 demonstrated that this protein was
localized between myoepithelial cells of the secretory duct in small puncta (**Supplementary Fig. S2**).
Taken together, these results indicate that Cx43 is the major gap junction constituent involved in the
segmental contraction of the secretory duct. These new data have been included in the revised
manuscript, page 9, line 251-page 10, line 280.

Reviewer #2:

*Line 118. "In the control condition with phosphate-buffered saline (PBS)"*

*--What is the concentration of extracellular calcium in these experiments (both control and agonist*
*conditions)? In the discussion the authors acknowledge a potential role for calcium (which makes*
*sense with this agonist and with actin-mediated contraction generally and in light of other work). It is*
*therefore important that physiological levels of extracellular calcium are present.*

We thank the reviewer for this comment. The PBS used in the experiments shown in **Fig. 3** was free
of divalent cations. Pilocarpine was dissolved in this $\text{Ca}^{2+}/\text{Mg}^{2+}$ -free PBS in these live imaging
experiments. Thus, pilocarpine-induced dislodgement of myoepithelial cells does not require the
extracellular milieu to be supplemented with divalent cations. The same experiment was also
performed in Krebs-Ringer bicarbonate solution (KRS) containing 1 mM CaCl_2 and 1.2 mM MgCl_2 ,
and, as was the case with PBS, pilocarpine induced dislodgement of myoepithelial cells in the KRS
(new **Fig. S1 and Supplementary movie 3**). No significant difference was observed in the magnitude
of dislodgement of Acti-stained myoepithelial cells in the experiments performed in PBS and in KRS.
The observation that pilocarpine was capable of inducing dislodgement of myoepithelial cells
irrespective of the presence or absence of divalent cations in the extracellular milieu has been included
in the revised manuscript, page 6, lines 125-127.

125 were also dislodged (**Fig. 3c**). The pilocarpine-induced dislodgement of myoepithelial
126 cells and luminal secretory cells was observed irrespective of the presence or absence of
127 divalent metal ions in the extracellular milieu (**Supplementary movie 3**). Furthermore,

Although pilocarpine induces contractile motions of secretory tubules in both the presence and the
absence of divalent cations in the medium, the live imaging experiments shown in **Figs. 4-8** and
**Supplementary movies 5-17** were performed in KRS containing divalent ions; this is partly because
of the seminal work by the late Professor Kenzo Sato on the contractility of secretory tubules of sweat
glands (Sato K, *Experientia* 1:631-633, 1977 [ref. 17]; Sato K *et al.*, *Am. J. Physiol.* 237:C177-C184,
1979 [ref.18]), in which he observed the contraction of secretory tubules in KRS. We apologize for
the insufficient and confusing description of the experimental conditions regarding the medium used
for live imaging of isolated sweat glands. We have revised the **Methods** section to clarify that PBS
was used as the medium in **Fig. 3** and **Supplementary movies 1, 2, and 4** and that we otherwise used
KRS containing divalent cations; this information appears on page 16, lines 477-479, in the revised
manuscript.

1-A: Nitta Gelatin, Inc., Osaka, Japan), which stably held the isolated eccrine glands. To
prevent the samples from drying, the mounted eccrine glands were immersed in Krebs-
Ringer bicarbonate solution (KRS) supplemented with 11 mM glucose at 25°C^{18} , **except**
**that the mounted eccrine glands were immersed in $\text{Ca}^{2+}/\text{Mg}^{2+}$ -free PBS in the experiments**
**shown in Fig. 3 and Supplementary movies 1, 2, and 4.**

*Lines 125-128. “Furthermore, the nuclei of myoepithelial cells without Acti-stain costaining showed*
*greater displacement after pilocarpine treatment (Fig. 3d, e; Supplementary movie 3), suggesting that*
*Acti-stain that directly binds to filamentous actin might interfere with the dynamic motion of actin*
*filaments in myoepithelial cells.”*

*--Movie S2 looks like obvious contraction. **Movie S3** actually looks like the sample has slipped with*
*agonist addition. If the authors are going to make this claim about the stain (influencing future use by*
*others), it should be supported by stronger evidence.*

We agree with the reviewer and have replaced **Movie S3** (renumbered as **Movie S4** in the revised
manuscript) with the one that displays the dynamic contractile motion of the nuclei of myoepithelial
cells.

*Lines 137-148. “Notably, contractile motions were more pronounced in the region where the eccrine*
*gland displayed concertina-like CMDR signals (Fig. 4d, red arrowheads). Because nerve fibers have*
*been shown to be wrapped around a part of the secretory coil, we reasoned that concertina-like signals*
*might arise from the sympathetic nerve fibers being wrapped around secretory ducts but not from*
*myoepithelial cells per se. To corroborate this possibility, we costained the secretory duct for α SMA*
*and PGP9.5, the latter of which is a marker of nerve fibers. PGP9.5-positive nerve fibers exclusively*
*surrounded the secretory ducts transversely, while α SMA- positive myoepithelial cells were*
*longitudinally aligned like thick stripes on the secretory duct (Fig. 4f). These results are consistent*
*with the possibility that concertina-like CMDR signals stem from nerve fibers and that myoepithelial*
*cells are directly governed by cholinergic nerve terminals.”*

*--This seems obvious. What hypothesis is being disproved here? The wording of this is also a bit*
*strange considering they have used a bath solution.*

In response to the reviewer’s criticism, we have revised and shortened the description on page 6, lines
137-148, in the original manuscript to eliminate the hypothesis-driven approach by simply referring
to our previous report (Kurata et al., *PLoS One* 12, 1-17, 2017 [ref. 5]) as follows:

*Notably, contractile motions were most pronounced in the region where the eccrine gland displayed*
*concertina-like CMDR signals (Fig. 4d, red arrowheads). Previously, we reported that PGP9.5-*
*positive nerve fibers were predominantly wrapped around the secretory portions of sweat glands⁵.*
*Consistent with this observation, double immunostaining for PGP9.5 and α SMA confirmed that the*
*PGP9.5-positive nerve fibers exclusively surrounded the secretory ducts transversely, while α SMA-*
*positive myoepithelial cells were longitudinally aligned like thick stripes on the secretory duct (Fig.*
*4f).*

The shortened description above has been included on page 6, lines 140-147, in the revised manuscript.

*Line 153-160. "Perspiration can also be induced via the activation of nicotinic acetylcholine receptors*
*in vivo. To assess the contribution of nicotinic stimulation to the contraction of the human eccrine*
*gland, we examined the effect of nicotine by our 3D live-tissue imaging. The nuclei did not move after*
*nicotine stimulation (Fig. 5c, d; Supplementary movie 7), indicating that muscarinic stimulation, but*
*not nicotinic stimulation, induces the contractile motion of the human eccrine gland as the basis for*
*sweat expulsion. Hereafter, we used muscarinic stimulation by pilocarpine to induce eccrine gland*
*contraction in vitro."*

*--The authors explore this but do not discuss further. If nicotinic stimulation causes contraction in vivo,*
*but this cannot be observed ex vivo, what is the difference? Is it a technical issue? This should be*
*discussed.*

We appreciate the reviewer's comment on this point. Perspiration can be induced not only by the
agonists of muscarinic acetylcholine receptors but also by an agonist of nicotinic acetylcholine
receptors, i.e., nicotine. Both muscarinic and nicotinic acetylcholine receptors are expressed in eccrine
sweat glands, as demonstrated by in situ hybridization and immunohistochemistry (Kurzen et al., 2004
[ref. 12]). Our results show that nicotinic stimulation does not trigger the contractile motion of the
secretory duct, making it likely that nicotinic stimulation governs sweat production from secretory
cells but not the contraction of myoepithelial cells. Consistent with this conclusion, Sato and Sato
(1981) reported that contraction of isolated eccrine sweat glands was induced by muscarinic agonists
(e.g., muscarine and pilocarpine) but not by nicotine [ref. 22]. They concluded that sweating in
response to nicotine was elicited through the axon reflex, as proposed by Coon and Rothman (1941;
[ref. 11]). We have included the above information in the revised manuscript (page 13, lines 378-388).

Perspiration can be induced not only by agonists of muscarinic acetylcholine receptors
but also by an agonist of nicotinic acetylcholine receptors, i.e., nicotine¹¹. Both
muscarinic and nicotinic acetylcholine receptors were detected in eccrine sweat glands
by *in situ* hybridization and immunohistochemistry¹². Our results show that nicotinic
stimulation does not trigger the contractile motion of the secretory duct, making it likely
that nicotinic stimulation governs sweat production from secretory cells but not the
contraction of myoepithelial cells. Consistent with this conclusion, Sato and Sato (1981)
reported that isolated eccrine sweat glands contracted in response to muscarinic agonists
(e.g., muscarine and pilocarpine) but not in response to nicotine²². They concluded that
sweating in response to nicotine was elicited through the axon reflex, as proposed by
Coon and Rothman¹¹.

*Paragraph line 162-178.*

*--I find this entire section unconvincing. Firstly, we are distinguishing only slight differences in the*
*staining patterns of a non-specific stain on live tissue. Secondly, the entire structure moves, making it*
*difficult to see where the contraction is initiated and where movement is active vs passive. The*
*mechanism of quantification cannot distinguish. This is also the case for the next section.*

In this paragraph, we aimed to provide evidence for glandular contraction of the secretory duct upon
pilocarpine stimulation. Because two regions, one with concertina-like staining due to surrounding
nerve fibers and the other with thick striped/less concertina-like staining, were discernible on the
secretory duct upon CMDR staining, we reasoned that glandular contraction would occur more
prominently in the concertina-like region. As noted by the reviewer's critique, there were only slight
differences in the CMDR staining patterns, and the entire structure moved after pilocarpine stimulation.
To overcome this hurdle and distinguish the contraction of the secretory duct along its length from the
movement of the entire structure, we measured the distance between a pair of fixed reference points
selected along the length of the same secretory tract, as shown in **Fig. 6b**. Using **Movie S9** as a
template, we measured the distance between such pairs of reference points selected from concertina-
like regions (n=11); we found that the secretory duct indeed was shortened by 20% 5 minutes after
pilocarpine addition and thereafter returned to the original length (**Fig. 6c**). The difference in length
was statistically significant (P<0.01). Such transient glandular contraction was not detected in the thick
striped/less concertina-like region, as shown in **Fig. 6d**. To strengthen this conclusion, we gathered
more data by choosing additional pairs of such reference points from **Movies S5** for both the
concertina-like (n=5) and thick striped/less concertina-like (n=5) regions (the **new Fig. S1**), and we
combined them with the data obtained from **Supplementary movies S5 and S9** to yield the **new Fig.**
**6c and 6d**, confirming the conclusion that the secretory duct contracted longitudinally in the
concertina-like region but not in the thick striped/less concertina-like region after pilocarpine
stimulation. We would like to stress that our quantification of the distance between two fixed reference
points on the same secretory tubules should be minimally affected by the contractile motion of the
entire structure.

*Line 188 "It should be noted that the displacement of the myoepithelial nuclei preceded that of the*
*luminal nuclei"*

*--what is the temporal resolution for these experiments? Can the authors comment on whether they*
*have the temporal resolution to quantify difference in the movement of one cell type and a cell type*
*that is physically connected and very close to it.*

The resolution of this time-lapse image recording was 30 or 40 seconds, depending on the sample
thickness. The luminal cells and the myoepithelial cells in the secretory duct differ in their nuclear
shapes; the nuclei of luminal cells are rounded, while those of myoepithelial cells are elliptical (**Fig.**
**2i**). Therefore, we were able to distinguish between luminal and myoepithelial cells on live imaging
based on their nuclear shapes. In **Fig. 6i**, the nuclear movements of both classes of cells in the
transverse section of the secretory duct were analyzed at 30-second intervals; it was observed that the
movement of myoepithelial cell nuclei preceded that of luminal cell nuclei by 60-180 seconds. We
revised the description of live imaging in the **Methods** section to clarify the temporal resolution in the
acquisition of time-lapse images (page 17, lines 486-487, in the revised manuscript).

0.75; Olympus Corporation) at intervals of 30 or 40 seconds for 30 minutes, depending
on the thickness of the samples. At each recording time, twelve to thirty Z slices at 2- μ m

*Line 197. The dominance of green signals in the frame at 30:00 was due to fluorescence quenching*
*after prolonged recording.*

*--Freely available imaging software (e.g., ImageJ) has bleach corrections tools that may be useful.*
*Have the authors inverted the colours between the movie and the figure. It appears like the magenta*
*is dominating in the movie.*

We appreciate the reviewer's comment and suggestion. **Fig. 6j** and **Supplementary movie S12** have
been corrected using the Bleach Correction tool in ImageJ. Accordingly, the above description in line
197 was deleted from the revised manuscript.

-Fig 8

--The IHC is very hard to make sense of. Is this actin staining or alpha smooth muscle actin staining. This would be much more clear with cell type specific stains, particularly in 2D images. For fixed tissue with IHC there is no reason not to use cell-type specific antibodies. This is essential to support their conclusions about cell type specific expression of the various connexin proteins.

We apologize for the low resolution of the IHC images shown in **Fig. 8d-g** in the original manuscript. In accordance with the reviewer's suggestion, we performed IHC using α SMA as a cell-type-specific marker for myoepithelial cells. The results have now been included as a **new Fig. 8e-j** in the revised manuscript, in which we costained tissue sections of eccrine glands for α SMA and the connexins detected at the transcript level in the myoepithelial cells. Thus, we sorted the myoepithelial cell-enriched CD29^{high}/CD49f^{high} subpopulation from the isolated eccrine gland and used qRT-PCR to explore which connexin members are expressed in the myoepithelial cells and which member has the highest expression. qRT-PCR analyses showed that myoepithelial cells expressed Cx26, Cx30, Cx31, Cx37, Cx43, and Cx45, of which Cx43 was most abundant (**new Fig. 8d**). In the **new Fig. 8e-j**, representative IHC images of Cx26, Cx30, Cx31, Cx43, and Cx45 are shown along with that of α SMA, a marker of myoepithelial cells. The results show that Cx43 and Cx45 were detected on α SMA-positive myoepithelial cells as puncta, although myoepithelial cells showed no such punctate signals for other connexins, including Cx26, Cx30, Cx31, and Cx37. Punctate signals were also detected for other connexins in the luminal cells of the secretory duct at variable densities. Notably, Cx43 signals were frequently localized between myoepithelial cells in small puncta (**new Supplementary Fig. S2**), raising the possibility that Cx43 is the major gap junction constituent involved in the segmental contraction of myoepithelial cells in the secretory duct. Given the predominance of Cx43 among gap junction proteins on myoepithelial cells, we also addressed the role of Cx43 in the pilocarpine-induced contractile motion of the secretory duct using gap junction inhibitors specific to Cx43, i.e., 2-aminoethoxydiphenylborane (2-APB) and Gap27. We found that both gap junction inhibitors blocked the pilocarpine-induced contractile motion as well as the nuclear displacement of the secretory duct (**new Fig. 8k; Supplementary movies 16 and 17**), corroborating the role of Cx43-containing gap junctions in the segmental contraction of myoepithelial cells in the secretory duct. These new results regarding the expression of connexins at both the RNA and protein levels in the eccrine gland as well as the inhibitory effects of 2-APB and Gap27 have been included in the revised manuscript (page 9, line 251–page 10, line 280, with **new Fig. 8d-j and Supplementary movies 16 and 17**).

*Fig. 8h*

*Can the authors confirm that the regions of human skin used were alternated and this isn't a region-*
 *specific phenomenon?*

In our *in vivo* sweating tests, we ran a pretest to visualize active sweat pores on the palm of the subjects
by the starch-iodine method (page 10, lines 285-286; page 19, lines 561-563). We selected two areas
showing no apparent difference in the density of active sweat pores to eliminate any bias reflecting
region specificity. Representative images of such pretests (*left*; boxed in *red*) and the tests with CBX
pretreatment (*right*) have been included in the **new Supplementary Fig. S3**.

284 **(Fig. 9a)**. Sweating was induced by autonomic stimulation by passive heating via a foot
285 bath. **Prior to the sweating test, we ran a pretest to visualize active sweat pores on the**
286 **palms of the subjects.** Palm sweating was locally inhibited by approximately 30% by

*Discussion*

*Lines 297-298*

*Reference these papers also showing myoepithelial contraction in 3D, which are highly relevant to*
*this study.*

*Lacrimal gland - <https://doi.org/10.1038/s41598-018-28227-x>*

*Mammary gland - <https://doi.org/10.1073/pnas.2016905117>*

One of the suggested papers, <https://doi.org/10.1038/s41598-018-28227-x>, has been cited as Ref. 24
in the original manuscript. Another paper on mammary gland imaging,
<https://doi.org/10.1073/pnas.2016905117>, has now been cited as Ref. 26 in the revised manuscript.

- 24. Hawley, D. *et al.* Myoepithelial cell-driven acini contraction in response to oxytocin receptor
stimulation is impaired in lacrimal glands of Sjögren's syndrome animal models. *Sci. Rep.* **8**,
9919 (2018). doi:10.1038/s41598-018-28227-x
25. Redman, R. S. Myoepithelium of salivary glands. *Microsc. Res. Tech.* **27**, 25–45 (1994).
26. Stevenson, A. J., Vanwallegem, G., Stewart, T.A. & Davis, F. M. Multiscale imaging of basal
cell dynamics in the functionally mature mammary gland. *Proc. Natl. Acad. Sci. U.S.A.* **117**,
26822-26832 (2020) doi:10.1073/pnas.2016905117

Reviewer #3:

*1. The authors did great work in the myoepithelial contraction - induced the peristaltic transport of*
*sweat. However, the authors did not provide enough evidence to show the role of gap junction in this*
*mediation work. The manuscript title emphasized the gap junction mediation which cannot be*
*convincingly supported by the data provided.*

We appreciate the reviewer's comment. Our conclusion that myoepithelial contraction is mediated by
gap junctions was primarily based on our observation that carbenoxolone (CBX), a gap junction
inhibitor, inhibited ex vivo contractile motions of the secretory duct of the sweat gland (**Fig. 8a-c**).
This conclusion was further supported by the inhibition of sweating on the palms by CBX (**Fig. 9a,**
**b**). However, CBX is known as a gap junction inhibitor that displays broad specificity toward
connexins and is also known to block voltage-gated Ca^{2+} channels and pannexin channels (Manjarrez-
Marmolejo & Franco-Perez, 2016 [ref. 16]). Therefore, the inhibition by CBX per se does not
convincingly support the involvement of gap junctions, as noted in the reviewer's critique. To
corroborate the involvement of gap junctions in myoepithelial contraction, we explored the expression
patterns of connexins in myoepithelial cells at the transcript level by qRT-PCR. Because myoepithelial
cells in the eccrine sweat glands have been characterized as a $\text{CD29}^{\text{high}}/\text{CD49}^{\text{high}}$ subpopulation
(Kurata et al., 2017 [ref. 5]), we dissociated the isolated eccrine glands with collagenase treatment and
obtained the myoepithelial cell-enriched $\text{CD29}^{\text{high}}/\text{CD49}^{\text{high}}$ subpopulation. qRT-PCR analyses of a
total of 17 connexins, namely, Cx26, Cx30, Cx30.3, Cx31, Cx31.1, Cx31.9, Cx32, Cx37, Cx40,
Cx40.1, Cx43, Cx45, Cx46, Cx47, Cx50, Cx58, and Cx62, showed that the connexins expressed in
the $\text{CD29}^{\text{high}}/\text{CD49}^{\text{high}}$ subpopulation were Cx26, Cx30, Cx31, Cx37, Cx43, and Cx45, of which Cx43
was most abundant (**new Fig. 8d**). Given the abundant expression of Cx43 in myoepithelial cells, we
addressed the role of Cx43 in the contractile motion of the secretory duct using gap junction inhibitors
specific to Cx43, i.e., 2-aminoethoxydiphenylborane (2-APB) and Gap27. Our results clearly showed
that both inhibitors strongly blocked pilocarpine-induced contractile motion and nuclear displacement
in the secretory duct (**new Fig. 8k**). Taken together, these results support the role of the gap junction
in the pilocarpine-induced segmental contractile motion of myoepithelial cells in the secretory duct.
These additional results, including qRT-PCR analyses and experiments using 2-APB and Gap27, are

now included in the revised manuscript (page 9, lines 251-261, and page 10, lines 272-280).

To further explore the role of gap junctions in the segmental contraction of the
secretory duct, we examined the expression levels of gap junction proteins, i.e., connexins
(Cx_s), in the myoepithelial cells of the secretory duct by quantitative reverse transcription
polymerase chain reaction (qRT-PCR). Because the myoepithelial cells in the eccrine
sweat glands have been characterized as a CD29^{high}/CD49^{high} subpopulation⁵, we
dissociated the isolated eccrine glands with collagenase and obtained the myoepithelial
cell-enriched CD29^{high}/CD49^{high} subpopulation. qRT-PCR analyses of a total of 17
connexins, namely, Cx26, Cx30, Cx30.3, Cx31, Cx31.1, Cx31.9, Cx32, Cx37, Cx40,
Cx40.1, Cx43, Cx45, Cx46, Cx47, Cx50, Cx58, and Cx62, showed that the connexins
expressed in the CD29^{high}/CD49^{high} subpopulation were Cx26, Cx30, Cx31, Cx37, Cx43,
and Cx45, of which Cx43 was most abundant (Fig. 8d). Immunofluorescence staining of

The role of Cx43 in the segmental contraction of myoepithelial cells was further
addressed by using 2-aminoethoxydiphenylborane (2-APB) and Gap27, which are gap
junction inhibitors specific to Cx43. 2-APB blocked the pilocarpine-induced contractile
motion of the secretory duct and the nuclear displacement of the luminal cells almost
completely (Fig. 8k red line; Supplementary movie 16). Gap27 also exhibited an
inhibitory effect, albeit less pronounced than that of 2-APB, on contractile motion and
nuclear displacement in the secretory duct (Fig. 8k gray line; Supplementary movie 17).
These results support the role of Cx43 in the pilocarpine-induced segmental contraction
of myoepithelial cells in the secretory duct.

*2. From the data provided by the authors Figure 8, we can see the blockage of gap junction impact*
*myoepithelial contraction, which indicated the potential role of gap junction on myoepithelial*
*contraction. Why to chose gap junctions? How about the tight junctions and anchoring junctions?*

We thank the reviewer for this comment. Myoepithelial cells contract upon neuronal or hormonal
stimulation in various exocrine secretory ducts, including the mammary, lacrimal, and salivary glands,
of which the mammary gland is best characterized with respect to the mechanism underlying
myoepithelial contraction (page 12, lines 342-352). Importantly, mammary myoepithelial cells have
been shown to communicate with each other through gap junctions (ref. 38; page 12, lines 342-352),
prompting us to investigate the involvement of gap junctions in myoepithelial contraction in sweat
glands. We have not yet examined other junctional complexes, including tight junctions and anchoring
junctions.

*3. Gap junctions are mainly controlled by the profile of connexin family. But the authors did not*
*correctly show the expression differences of Connexin family. Why to choose Cx26, Cx30, Cx32 and*

*Cx43 in myoepithelial cell? How many members of connexin family are expressed in myoepithelial*
*cell? Which member has the highest expression in myoepithelial cell and which member plays the most*
*important role? From Figure 8d-g, we cannot judge these information.*

We wish to thank the reviewer for this comment. We fully agree with the reviewer in that we need to
understand how many members of the connexin family are expressed in the myoepithelial cells of the
eccrine sweat gland and which member has the highest expression and plays the most important role.
In the original manuscript, we chose Cx26, Cx30, Cx32, and Cx43 because these connexins are
reportedly expressed in the human epidermis and mammary glands (Salomon et al., *J. Invest.*
*Dermatol.* **103**, 240–247 (1994); refs. 38, 39). Given the comments by the reviewer, we examined
the expression of the connexin family in the myoepithelial cells of the eccrine sweat gland as detailed
above (*the responses for the reviewer's comment 1*). Our results clearly show that Cx43 is the most
abundant connexin expressed in the myoepithelial cells of the sweat gland and is involved in the
segmental contraction of myoepithelial cells, as evidenced by the fact that the pilocarpine-induced
contractile motion of the secretory duct was inhibited by 2-APB and Gap27, two gap junction
inhibitors specific to Cx43. These additional results are included in the revised manuscript, page 9,
lines 251-261, and page 10, lines 272-280.

To further explore the role of gap junctions in the segmental contraction of the
secretory duct, we examined the expression levels of gap junction proteins, i.e., connexins
(Cxs), in the myoepithelial cells of the secretory duct by quantitative reverse transcription
polymerase chain reaction (qRT-PCR). Because the myoepithelial cells in the eccrine
sweat glands have been characterized as a CD29^{high}/CD49^{fhigh} subpopulation⁵, we
dissociated the isolated eccrine glands with collagenase and obtained the myoepithelial
cell-enriched CD29^{high}/CD49^{fhigh} subpopulation. qRT-PCR analyses of a total of 17
connexins, namely, Cx26, Cx30, Cx30.3, Cx31, Cx31.1, Cx31.9, Cx32, Cx37, Cx40,
Cx40.1, Cx43, Cx45, Cx46, Cx47, Cx50, Cx58, and Cx62, showed that the connexins
expressed in the CD29^{high}/CD49^{fhigh} subpopulation were Cx26, Cx30, Cx31, Cx37, Cx43,
and Cx45, of which Cx43 was most abundant (**Fig. 8d**). Immunofluorescence staining of

The role of Cx43 in the segmental contraction of myoepithelial cells was further
addressed by using 2-aminoethoxydiphenylborane (2-APB) and Gap27, which are gap
junction inhibitors specific to Cx43. 2-APB blocked the pilocarpine-induced contractile
motion of the secretory duct and the nuclear displacement of the luminal cells almost
completely (**Fig. 8k** red line; **Supplementary movie 16**). Gap27 also exhibited an
inhibitory effect, albeit less pronounced than that of 2-APB, on contractile motion and
nuclear displacement in the secretory duct (**Fig. 8k** gray line; **Supplementary movie 17**).
These results support the role of Cx43 in the pilocarpine-induced segmental contraction
of myoepithelial cells in the secretory duct.

4. Besides, there is no other data to explain the underlying regulation mechanism, which is the big
defect of the manuscript.

As detailed above, we gathered more data on the involvement of gap junctions in the pilocarpine-
induced contractility of myoepithelial cells by characterizing the connexins expressed in myoepithelial
cells; using gap junction inhibitors specific to Cx43, we demonstrated that this protein, the most
abundantly expressed connexin in myoepithelial cells, is involved in the pilocarpine-induced
contractile motion of myoepithelial cells. These new results, together with the results obtained through
the *in vivo* perspiration test, in which carbenoxolone suppressed sweat secretion induced by passive
heating, support the role of gap junctional intercellular communication (GJIC) in the contractile
function of myoepithelial cells in sweat secretion from human eccrine glands, highlighting the role of
Cx43-containing gap junctions. The revised manuscript discusses the role of GJIC in myoepithelial
contraction during sweat secretion with the new data described above (page 13, line 390 through page
14, line 417).

Page 15, line 422, “Acti-stain” Writing errors.

According to the reviewer’s suggestion, we confirmed the wording in line 422. Acti-stain™ 488 is a
tradename for Alexa 488-conjugated phalloidin, which is available from Cytoskeleton, Inc., Denver,
USA.

Reviewers' comments:

Reviewer #2 (Remarks to the Author):

I thank the authors for their additional work in addressing the reviewers' comments.

I have a few remaining (major) concerns about these new data.

1. Throughout the rebuttal and the revised manuscript, the authors reference their use of 2APB as a "specific Cx43 blocker". 2APB has complex pharmacology and is anything but specific for Cx43. Depending on concentration (which I could not locate in the manuscript) and conditions, 2APB can block or enhance store-operated calcium entry, and inhibit IP3Rs, TRP channels etc. Its inhibition of pilocarpine induced contraction is likely confounded by inhibition of ER release and its presentation and interpretation in this manuscript needs to be completely revisited.

Moreover, all pharmacological agents used must be clearly described (concentration, incubation times etc).

2. I cannot quite comprehend from the author's explanation why initial experiments were performed in calcium free medium. If contractions happen spontaneously in calcium containing buffer, how did this affect all subsequent findings examining contraction (done in 1 mM Ca)?

Some in-depth discussion is needed in how they think their system operates irrespective of extracellular calcium (with pilocarpine stimulation) and how/why this would be different to "spontaneous" contractions in the absence of pilocarpine stimulation

3. To my question about the timing between myoepithelial contraction and displacement of the immediate underlying cell layer (luminal) cells: I understand their method of segmentation of the two populations, however I am confused by the response that there is a 60-180 s delay.

How is it physiologically possible for such a long delay between the contraction of one population and the movement of the immediately adjacent cell type. Have I misunderstood what is being measured here? If the authors stand by this finding, it needs extensive discussion.

Reviewer #3 (Remarks to the Author):

The author answered my previous question very carefully; I am satisfied with these responses.

| Reviewer #2 (Remarks to the Author):

| *I thank the authors for their additional work in addressing the reviewers' comments.*

| *I have a few remaining (major) concerns about these new data.*

| *1. Throughout the rebuttal and the revised manuscript, the authors reference their use*
*of 2APB as a "specific Cx43 blocker". 2APB has complex pharmacology and is*
*anything but specific for Cx43. Depending on concentration (which I could not locate in*
*the manuscript) and conditions, 2APB can block or enhance store-operated calcium*
*entry, and inhibit IP3Rs, TRP channels etc. Its inhibition of pilocarpine induced*
*contraction is likely confounded by inhibition of ER release and its presentation and*
*interpretation in this manuscript needs to be completely revisited.*

| We thank the reviewer for reminding us that 2-APB is not an inhibitor specific for Cx43.
| We agree with the reviewer, being aware of the complex pharmacology of 2-APB, which
| blocks Cx43-mediated gap junctional peptide transfer (Neijssen J, *et al.* Nature 434:83-
| 88, 2005 [ref. 17]) but also inhibits IP3Rs and TRP channels and has a dose-dependent
| bimodal effect on store-operated calcium entry (Prakriya & Lewis, Physiol Rev 95:1383-
| 1436, 2015 [ref. 19]). Thus, the inhibitory effect of 2-APB on pilocarpine-induced
| myoepithelial contraction is likely confounded by the complex pharmacology of 2APB.
| Accordingly, we significantly revised the description of the inhibitory effect of 2-APB on
| the pilocarpine-induced contractile motion of isolated eccrine glands by referring to the
| complex pharmacology of 2-APB, including the dose-dependent bimodal effect on store-
| operated calcium entry (page 10, lines 274-286, in the revised manuscript).

| *Moreover, all pharmacological agents used must be clearly described (concentration,*
*incubation times etc).*

| Information regarding all pharmacological agents, such as concentrations and incubation
| times, has been included in the revised manuscript (page 18, lines 507-509). 2-APB was
| used at 50 μ M to inhibit the pilocarpine-induced contractile motion of the eccrine glands
| (page 18, line 508).

| *2. I cannot quite comprehend from the author's explanation why initial experiments*
*were performed in calcium free medium. If contractions happen spontaneously in*

*calcium containing buffer, how did this affect all subsequent findings examining*
*contraction (done in 1 mM Ca)?*

*Some in-depth discussion is needed in how they think their system operates irrespective*
*of extracellular calcium (with pilocarpine stimulation) and how/why this would be*
*different to "spontaneous" contractions in the absence of pilocarpine stimulation.*

We apologize for the confusing description on page 6, lines 119-121 (posted below),
regarding the control condition with PBS.

119 pseudocolored green or magenta at different time points (Fig. 3a). **In the control condition**
120 **with phosphate-buffered saline (PBS), the spontaneous motion of the Acti-stained**
121 **myoepithelial cells was rarely observed**, leaving the colors almost completely merged in
122 *white* for both myoepithelial actin structures and nuclei (Fig. 3b; **Supplementary movie**
123 **1**). Upon pilocarpine stimulation, however, spindle-shaped myoepithelial cells responded

What we intended here was simply that the spontaneous motion of Acti-stained
myoepithelial cells was rarely observed in the control condition, i.e., in the absence of
pilocarpine. We did not intend that spontaneous motion was observed in the presence of
Ca but not in the absence of Ca. Spontaneous contractile motion was rarely observed
without pilocarpine stimulation irrespective of the presence or absence of Ca. Thus, we
started recording the contractile motion of the isolated eccrine gland approximately 10
minutes before pilocarpine addition, thereby confirming that spontaneous contraction
did not occur with the gland tested (see **Figs. 5-8** and page 18, lines 501-504).

In our initial experiments on ex vivo imaging of isolated sweat glands, we used Acti-
stain 488, a fluorochrome-conjugated phalloidin, as a live stain. However, phalloidin
has been reported to induce cell death of hepatocytes in the presence of calcium (Kane
AB, *et al*, Proc Natl Acad Sci USA, 77:1177-1180, 1980), which prompted us to use
calcium-free PBS as a medium to avoid cell damage during live imaging with Acti-
stain. This was the reason we employed calcium-free medium in our initial experiments.
In subsequent experiments, however, we found that Acti-stain compromises pilocarpine-
induced contractile motion of the isolated sweat glands (**Fig. 3d, e**) and therefore is not
suitable as a live stain to study the contractile motion of sweat glands. We therefore
employed CellMask Deep Red (CMDR), a cell membrane stain that does not interfere
with actin dynamics, for live imaging of sweat gland contraction. Because CMDR live
staining is compatible with calcium-containing Krebs-Ringer bicarbonate solution
(KRS), we employed KRS, which offers a more physiological milieu than calcium-free
PBS and has been used in seminal work by Professor Kenzo Sato on the contractility of

secretory tubules of sweat glands (Sato K, *Experientia* 1:631-633, 1977 [ref. 20]; Sato K
*et al.*, *Am. J. Physiol.* 237:C177-C184, 1979 [ref. 21]).

The reason why isolated sweat glands contract upon pilocarpine stimulation in the
absence of Ca remains to be addressed. Concepcion *et al.* (*JCI* 126:4303-4318, 2016; [ref.
33]) reported that acetylcholine (Ach) stimulation of isolated mouse sweat glands induced
comparable calcium release from ER stores in wild-type and Orai1-/Stem1-deficient mice
immediately after Ach stimulation. Because intracellular calcium was not depleted by
EGTA in our experimental conditions, such initial calcium release from ER stores by
pilocarpine stimulation may be sufficient for the induction of contractility of
myoepithelial cells in our experimental conditions. We have discussed our observation
that the secretory duct of isolated sweat glands contracts irrespective of the presence or
absence of calcium in the medium on page 13, lines 356-362 in the revised manuscript.

*3. To my question about the timing between myoepithelial contraction and displacement*
*of the immediate underlying cell layer (luminal) cells: I understand their method of*
*segmentation of the two populations, however I am confused by the response that there*
*is a 60-180 s delay.*

*How is it physiologically possible for such a long delay between the contraction of one*
*population and the movement of the immediately adjacent cell type. Have I*
*misunderstood what is being measured here? If the authors stand by this finding, it*
*needs extensive discussion.*

We thank the reviewer for pointing out this issue. In our previous response to the
reviewer's comment, we answered that "the movement of myoepithelial nuclei preceded
that of luminal cell nuclei by 60-180 seconds". The time lag of 60-180 seconds was
based on our observation that the movement of luminal cell nuclei was usually detected
2-6 frames behind the movement of myoepithelial nuclei when the movie was recorded
at intervals of 30-40 seconds. However, we did not pay enough attention to locating the
luminal cells adjacent to the myoepithelial cells of interest; instead, we focused on
luminal cells whose nuclear movement was easy to follow even when they were distant
from the myoepithelial cells (see the original **Fig. 6g**). Given the reviewer's comment,
we reexamined the movement of the nuclei of the myoepithelial cells and those of
adjacent and distant luminal cells, designated Lum1 and Lum2 in **new Fig 6g**, on four
consecutive frames where initial displacement of their nuclei was detected after
pilocarpine stimulation. The results are shown in the **new Fig. 6g**. We found that the

nuclei of the selected myoepithelial cell and that of the adjacent luminal cell moved
almost simultaneously without a delay of more than two frames. Accordingly, we
gathered more data by analyzing a total of six pairs of myoepithelial and luminal cells
adjacent to each other from two independent movies, confirming that the nuclei of the
myoepithelial cells and those of the adjacent luminal cells move almost simultaneously.
The results are shown in the **new Fig. 6i**. We again thank the reviewer for bringing this
issue to our attention. These new results have been noted on page 8, lines 192-195, in
the revised manuscript.

REVIEWERS' COMMENTS:

Reviewer #3 (Remarks to the Author):

The topic is interesting and the authors provide solid evidence to support their conclusions. This work emphasizes the role of CX43, which is originally used as a switch for intercellular gap junction communication, on the permanent transport of sleep in human eccrine lands, and brings new understanding in the transportation path of sweat.